# AUTOMATED CAPABILITY DISCOVERY VIA FOUNDATION MODEL SELF-EXPLORATION

**Cong Lu**[*,1,2]
conglu97@outlook.com

**Shengran Hu**[*,1,2]
srhu@cs.ubc.ca

**Jeff Clune**[1,2,3]
jclune@gmail.com

[*]Equal Contribution [1]University of British Columbia
[2]Vector Institute [3]Canada CIFAR AI Chair

## ABSTRACT

Foundation models have become general-purpose assistants, exhibiting diverse capabilities across numerous domains through training on web-scale data. It remains challenging to precisely characterize even a fraction of the full spectrum of capabilities and potential risks in any new model. Existing evaluation approaches often require significant human effort, and it is taking increasing effort to design ever harder challenges for more capable models. We introduce AUTOMATED CAPABILITY DISCOVERY (ACD), a framework that designates one foundation model as a *scientist* to systematically propose open-ended tasks probing the abilities of a *subject* model (potentially itself). By combining frontier models with ideas from the field of open-endedness, ACD automatically and systematically uncovers both surprising capabilities and failures in the subject model. We demonstrate ACD across a range of foundation models (including the GPT, Claude, and Llama series), showing that it automatically reveals thousands of capabilities that would be challenging for any single team to uncover. We further validate our method's automated scoring with extensive human surveys, observing high agreement between model-generated and human evaluations. By leveraging foundation models' ability to both create tasks and self-evaluate, ACD is a significant step toward scalable, automated evaluation of novel AI systems. All code and evaluation logs are open-sourced at https://github.com/conglu1997/ACD.

## 1 INTRODUCTION

Large Language Models (LLMs; Gemini Team, 2024; OpenAI, 2024b; Touvron et al., 2023), trained on internet-scale datasets, have revolutionized natural language processing by demonstrating strong general-purpose capabilities. These "Foundation Models" (FMs; Bommasani et al., 2021) display exceptional performance on tasks requiring common-sense knowledge (Talmor et al., 2019), reasoning (Wei et al., 2022), and comprehension (Chang et al., 2024), enabling applications ranging from conversational agents (Brown et al., 2020) to code generation (Gauthier, 2024). Recently, agentic systems powered by foundation models have even shown the capacity to propose and investigate new scientific ideas (Lu et al., 2024b) and provide ever-better agentic systems (Hu et al., 2024). However, identifying unknown abilities or failure modes in FMs remains a major challenge, especially because such knowledge is crucial to ensuring both safe deployment and maximizing real-world utility.

Traditional evaluation techniques—centered around human-created benchmarks (BIG-bench authors, 2023; Chen et al., 2021a; Cobbe et al., 2021; Dua et al., 2019; Hendrycks et al., 2021a;b; Phan et al., 2025; Zellers et al., 2019)—are labor-intensive to create and limited by predefined categories, often failing to capture the full spectrum of a model's capacities or unearthing surprising new behaviors pre-deployment. Moreover, as models become more advanced, they may saturate or overfit these benchmarks, so those metrics may not reflect broader performance gains. Users also commonly encounter unique use cases and failure modes not covered by benchmarks. While frequently updating or creating new test suites (Phan et al., 2025; White et al., 2024) attempts to address these issues, continually devising new tasks is expensive, not model-specific and will fail to probe the 'unknown unknowns' (things that benchmark creators do not think to include). This underscores the need for

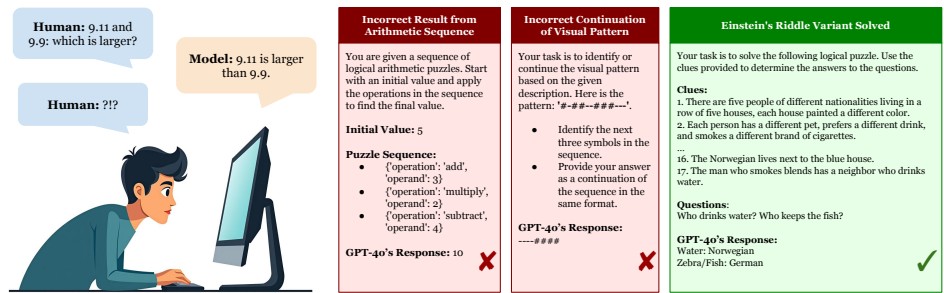

Figure 1: **(Left)** Humans typically evaluate novel foundation models through trial and error, alongside benchmarks. They often discover new surprising capabilities or failures: like counting how many "r"s are in "strawberry" or identifying which is bigger, 0.9 or 0.11. **(Center and Right)** AUTOMATED CAPABILITY DISCOVERY (ACD) mirrors human evaluation efforts by using a *scientist* model to automatically discover and assess the capabilities of a *subject* model in an open-ended manner. Illustrated here are two **surprising failures** (the model fails to perform three arithmetic operations in sequence, and fails to correctly continue a symbol pattern with '###') and a selected **success** (the model successfully solves a variant of Einstein's riddle with 17 clues) uncovered by ACD on GPT-4o. See Appendix E.3 for more examples. Models are evaluated using chain-of-thought (Wei et al., 2022).

scalable, efficient evaluation methods that are cheap and require minimal overhead to keep pace with rapidly evolving foundation models (Bowman et al., 2022).

We introduce AUTOMATED CAPABILITY DISCOVERY (ACD), a framework that augments existing evaluation approaches by automating the discovery of a foundation model's capabilities and failure modes. It designates one model as a *scientist* to systematically propose open-ended tasks for a *subject* model, which could be itself or a different foundation model (Section 4). Concretely, ACD instructs the scientist to propose *interesting new* challenges (Faldor et al., 2024; Lu et al., 2024b; Pourcel et al., 2024b; Shah et al., 2024a; Zhang et al., 2024a;b), asks the subject to attempt them, and evaluates performance (Zheng et al., 2023), all automatically. This mirrors how humans might try everything from their favorite model gotcha questions to new challenging problems when exploring a new model—though with ACD, the model takes on the role of evaluator. By removing manual task design from the process, ACD can automatically and relatively inexpensively expose a wide range of strengths and weaknesses in the subject model.

We demonstrate ACD on several foundation models, including GPT-4o (OpenAI, 2024b), Claude Sonnet 3.5 (Anthropic, 2024), and Llama3-8B (Llama Team, 2024) (Section 5). We show that ACD uncovers a large variety of capabilities, ranging from elementary arithmetic tasks to complex puzzle solving, resulting in thousands of automatically discovered tasks. Many tasks illustrate useful model capabilities, such as multi-step reasoning and structured workflows, whereas others reveal surprising failure modes that would seem trivial to humans (Figure 1). We provide numerous examples in our evaluations, spanning cryptography, code generation, memory-based logic, advanced mathematics, legal queries, puzzle design, and creative writing (Appendix E.3). To validate ACD's automated task generation and scoring, we conduct large-scale human surveys on tasks discovered by GPT-4o, showing high rates of tasks being deemed valid and agreement between the model's self-evaluation and human judgments (Section 5.1). Furthermore, ACD automatically compiles a concise **"Capability Report"** of discovered capabilities and failure modes (Section 6), enabling quick inspection and easier dissemination of results or flagging issues pre-deployment (Section 7).

By harnessing the capacity of foundation models to self-assess, ACD paves the way for scalable, automated evaluation of these models. It can help systematically identify emergent and potentially concerning behaviors before real-world deployment. As foundation models continue to advance, techniques like ACD will be crucial to align their development with human values and ensure responsible use by uncovering beneficial and risky behaviors before real-world deployment. Finally, ACD could enable models to generate interesting challenges for themselves to learn on, potentially driving self-improvement in the future (Clune, 2019; Faldor et al., 2024).

## 2 BACKGROUND

**Open-ended Discovery Algorithms.** Open-ended algorithms (Stanley & Lehman, 2015; Stanley et al., 2017) aim to continuously generate novel and diverse artifacts (Hughes et al., 2024) within a

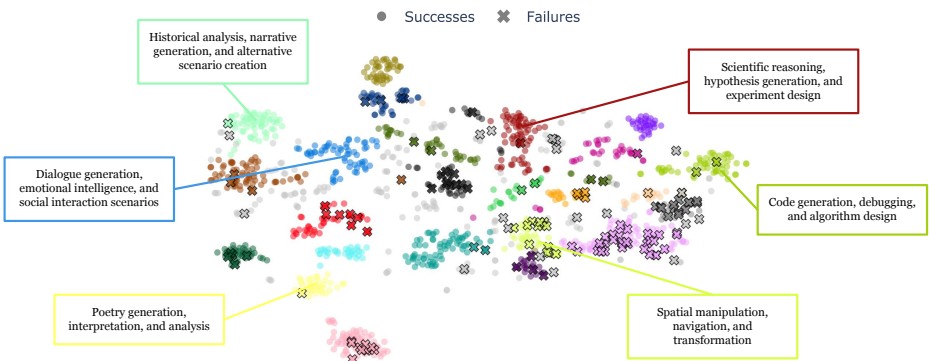

Figure 2: Task families discovered by AUTOMATED CAPABILITY DISCOVERY on GPT-4o (serving as both *scientist* and *subject*) over 5000 generations. Each point represents one of the 1330 tasks that passed the "interestingly new" filter, visualized in 2D via t-SNE. ACD enables GPT-4o to *self-discover* diverse capabilities and failure modes, with tasks that cluster into **25** high-level categories (different colors, listed in Appendix E.1), spanning *puzzle-solving, code generation, scientific reasoning, creative writing, and legal interpretation*. See Sections 4 and 5.1 for full details, and Appendix E.3 for selected examples.

search space, rather than focusing on a fixed objective. These algorithms emulate human creativity by autonomously exploring new artifacts, increasingly supported by large foundation models that can encode intrinsic notions of "interestingness" (Faldor et al., 2024; Lu et al., 2024c; Zhang et al., 2024a). They have been applied to evolving novel robot morphologies in code (Lehman et al., 2022), generating new reinforcement learning environments (Faldor et al., 2024; Wang et al., 2019; 2020), discovering novel loss functions (Lu et al., 2024a) and agentic systems (Hu et al., 2024), and investigating scientific hypotheses (Lu et al., 2024b).

Generally, these algorithms maintain and update an archive $\mathcal{A}$ of discovered artifacts. At iteration $t$, they sample a new artifact $a_t$ from a foundation model $M$ conditioned on a subset $C_{t-1}$ of previously discovered artifacts, typically limited in size for computational feasibility. The generated artifact $a_t$ is evaluated for novelty (e.g., via embedding-based similarity), and then added to the archive if sufficiently different from others in $\mathcal{A}$. ACD adapts these principles to systematically reveal a foundation model's capabilities, treating each novel *capability or failure* as a generated "artifact".

## 3 RELATED WORK

**Open-Ended Discovery with Foundation Models.** The field of open-endedness (Stanley, 2019) aims to continually discover diverse and novel artifacts forever. Recent methods leverage the generative capabilities and vast prior knowledge of foundation models (FMs) to accelerate this process (Faldor et al., 2024; Hu et al., 2024; Lehman et al., 2022; Zhang et al., 2024a) by harnessing a foundation model's intrinsic notion of interestingness (Faldor et al., 2024; Hu et al., 2024; Lu et al., 2024c; Zhang et al., 2024a) to construct the next proposal, analogous to human innovation. Notable examples include ELM (Lehman et al., 2022) which evolves novel robot morphologies; OMNI-EPIC (Faldor et al., 2024), which automatically designs novel environments for reinforcement learning (RL) agents; DiscoPOP which discovers new loss functions for preference optimization algorithms (Lu et al., 2024a); ADAS (Hu et al., 2024), which evolves novel designs for LLM-based agentic systems; and The AI Scientist (Lu et al., 2024b), which seeks to automate the entire scientific process by proposing novel ideas, conducting experiments, and writing a scientific paper summarizing the results.

**Automated Evaluation of Foundation Models.** Recent research also investigates automated evaluation of FMs, moving beyond static, human-designed test suites. Rainbow Teaming (Samvelyan et al., 2024) applies Quality-Diversity algorithms (Mouret & Clune, 2015; Pugh et al., 2016) to find novel adversarial attacks that stress-test FMs for safety. Similarly, Jiang et al. (2024); Pavlova et al. (2024); Zheng et al. (2024); Zhou et al. (2024) automate the red teaming (probing a system for weaknesses) process. These works expand the comprehensiveness of existing safety checks but do not have the ability to create new tasks that might reveal completely unknown FM capabilities. Other techniques generate new debate topics and evaluate FMs through multi-round debate between them (Zhao et al., 2024), discover open-ended programming challenges (Pourcel et al., 2024a), or devise visual recognition and reasoning tasks from a collection of visual assets (Zhang et al., 2024b).

Meanwhile, Shah et al. (2024b) produces challenging math problems from existing datasets and human-in-the-loop supervision. However, the generated tasks in these works tend to focus on a restricted domain, which fails to provide an overview of a model's abilities across a wide array of skills and limits the discovery of surprising capabilities of FMs.

## 4   AUTOMATED CAPABILITY DISCOVERY

Given a foundation model we wish to evaluate (the *subject*), AUTOMATED CAPABILITY DISCOVERY (ACD) designates another foundation model as a *scientist* to propose new tasks and then evaluate how well the subject model performs. The scientist and subject could be the same model or different, but in either case, they are both foundation models, so we refer to this as "foundation model self-exploration." By iteratively refining tasks to uncover interesting or surprising outcomes, ACD aims to automate much of the process of revealing a model's capabilities. Below, we outline the key stages of ACD. (*See Appendix B for the full* ACD *prompts.*)

### 4.1   DEFINITION OF TASK FAMILIES

We adopt a simplified version of the METR Task Standard (METR Task Standard Team, 2024), an established format for packaging tasks to evaluate foundation models. In particular, ACD instructs the scientist to define and generate broad "task families" as a systematic way to cover entire categories of capabilities—ranging from simple

Table 1: Example metadata for a simple "Hello World" task family.

| Key | Value |
|---|---|
| name | hello_world |
| description | return a greeting string |
| capability being measured | basic string manipulation |

knowledge recall to more complex reasoning or coding. Each family has metadata which includes a name, a description, and the exact capability being measured. Table 1 shows an example of how such metadata is seeded for a trivial "Hello World"-style string repetition task.

We leverage the LLM's coding abilities to translate high-level task descriptions into Python code that can be automatically evaluated. Each task family (METR Task Standard Team, 2024) is structured with:

1. **Specific Task Instances**: Subtasks are generated with unique data to probe different nuances of the same capability.
2. **Instruction Provision**: Each subtask includes instructions for the subject model.
3. **Scoring Mechanism**: A programmatic check for tasks with a single correct answer, or a GPT-4o-based judge (Zheng et al., 2023) if the task requires more open-ended judgment (Appendix A.2).

Appendix A.1 shows a full code snippet for the "Hello World" example in Table 1. This task family may include the strings "Hello, world!" or "Greetings, universe!" as subtasks, the instructions to the subject model may be "Please repeat the following message exactly as it is: {...}", and the scoring mechanism may be an exact string comparison. For more open-ended tasks, we demonstrate that using foundation models as open-ended automated judges can work, since often it is easier to recognize the successful solution to a particular task than generate one.

### 4.2   GENERATING TASKS

Following principles from the field of open-endedness (Section 2), ACD operates in a loop:

1. **Maintain an Archive:** An archive (Lehman & Stanley, 2011; Mouret & Clune, 2015) of tasks that have been discovered thus far is kept. It is seeded with trivial tasks (like those in Section 4.1). At each iteration, the scientist sees a randomly sampled subset of these tasks as context.
2. **Propose a New Task Family:** The scientist proposes a new task family (written in Python code), using chain-of-thought (Wei et al., 2022) and self-reflection (Shinn et al., 2023) to catch errors (Appendix B). During self-reflection, the scientist also checks how easily the subject solves the current task family and adapts difficulty accordingly.
3. **Filter for Novelty:** The scientist discards proposals that overlap too closely with existing tasks, by considering whether the task is "interestingly new" (Zhang et al., 2024a) with respect to its nearest neighbors computed via `text-embedding-3-small` (OpenAI, 2024c) (Appendix B.3).

4. **Test the Subject Model:** The subject attempts these tasks using chain-of-thought (Appendix B.2) as a lightweight way to elicit greater capabilities from the FM. The scientist uses $n$-shot evaluation to robustly score each task. All completed tasks are stored in the archive, logged as "discovered capabilities" when consistently solved or "failure modes" when consistently failed.

We can repeat these steps for thousands of iterations until sufficiently many task families have been discovered.

## 5 EMPIRICAL EVALUATION

We now demonstrate ACD's performance in discovering diverse capabilities across several foundation models, including GPT-4o (OpenAI, 2024b), Claude Sonnet 3.5 (Anthropic, 2024), and Llama3-8B (Llama Team, 2024). First, we provide an in-depth examination of GPT-4o acting as both scientist and subject, followed by experiments with different scientist-subject pairings and cross-model analyses. We run our algorithm for 5000 generations for all evaluations. Further details on hyperparameters and evaluation protocols appear in Appendices C and G.

### 5.1 CASE STUDY AND HUMAN EVALUATION ON GPT-4O

We begin by analyzing ACD with GPT-4o serving as both scientist and subject. In Figure 2, we visualize all discovered tasks by embedding each task's description in a 2D t-SNE (Van der Maaten & Hinton, 2008) plot, grouped by HDBSCAN (McInnes et al., 2017). From these 5000 generations, we discover 1330 interestingly new tasks, which fall into 25 distinct clusters (Table 4 in Appendix E.1). The tasks span wide-ranging areas, including puzzle-solving and creation (e.g., Sudoku, logic riddles, custom word puzzles), code generation and debugging, advanced math, creative writing, and legal text interpretation. We provide many examples from our evaluations, spanning cryptography, code generation, memory-based logic, advanced mathematics, legal queries, puzzle design, and creative writing in Appendix E.3.

**Surprising Spatial Reasoning**

**Instructions:**
Your task is to visualize and describe the 3D object based on 2D projections. Projections:

- Front view: A square base with a triangular top. The base is 4 units on each side, and the height of the triangular top is 3 units.
- Side view: A rectangle with a triangular top. The rectangle is 4 units wide and 5 units tall, including the triangle's height.
- Top view: A square with 4 units on each side.

**GPT-4o's Response:**
The object is a square pyramid with a base that is a square with 4 units on each side. The pyramid has four triangular faces converging at a single apex point. The height of the pyramid from the base to the apex is 3 units. The total height of the object, including the base and the apex, is 5 units. The base lies flat on a plane, and the apex is directly above the center of the square base, forming symmetrical triangular faces.

**Unable to Follow Simple Instructions in Sequence**

**Instructions:**
Your task is to perform the following text transformations on the given input text: "The quick brown fox jumps over the lazy dog."

Transformations:
1. Reverse the order of words
2. Capitalize the first letter of each word
3. Replace 'dog' with 'cat'

**GPT-4o's Response:**
Dog. Lazy The Over Jumps Fox Brown Quick The

Here, we examine a few in detail. Figure 1 (right) highlights three surprising tasks discovered by ACD that reveal GPT-4o sometimes fails at seemingly trivial operations. For instance, it incorrectly computes a sequence of three arithmetic operations and fails to continue a simple pattern for three symbols as requested. On the other hand, it correctly solves a novel variant of Einstein's riddle (that we could not find online) with 17 logical clues, demonstrating unexpectedly strong multi-step reasoning. We show two further discovered tasks to the left, which illustrate a surprisingly good spatial reasoning success **(left box)** and a puzzling failure to follow basic, multi-step instructions **(right box)**. These discovered tasks show that ACD can *entirely autonomously* pinpoint subtle failure modes that GPT-4o exhibits, just as human evaluators might, at a low cost of approximately $0.1 per task created (Appendix C.1). Such insights could strongly inform future training or target fixes prior to user deployment.

**Human Evaluation.** To verify that these generated tasks are coherent and to assess the accuracy of automated scoring at scale, we conducted a human survey via CloudResearch (details in Appendix G). Participants were shown each task's short description, the exact instructions, and the subject model's final submission. They then evaluated whether the task was valid and coherent, estimated the difficulty on a five-point scale ("Very Easy" through "Very Difficult"), and judged whether the model's submission was correct.

Figure 3(a) shows that 92.2% of tasks were considered clear and valid by human participants—that is, the instructions were judged to align well with the task description and the capability being measured.

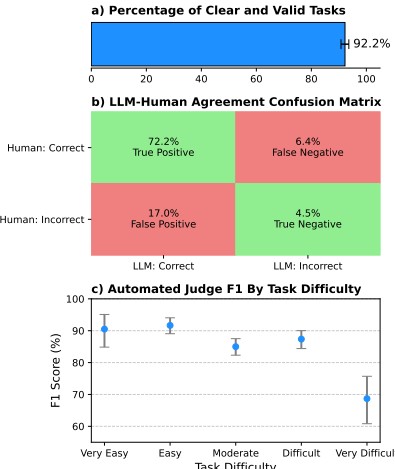

Figure 3: Human evaluation results for tasks discovered by ACD when GPT-4o serves as both *scientist* and *subject* (Section 4). (a) Most automatically generated tasks were judged clear and valid. (b) The confusion matrix shows that automated scoring largely agrees with human judgments, but exhibits a slight positive bias (more false positives than false negatives). The F1 score of LLM-Human agreement is quite high at 0.86. (c) The automated judge's F1 is reasonably high for tasks rated by humans as "Very Easy" to "Difficult," though it drops on "Very Difficult" tasks. Error bars are 95% bootstrapped confidence intervals across generated task families.

Additionally, as seen in Figure 3(b), the automated scoring largely agrees with human judgments (with an F1 score of 0.86). We observe a slight positive bias from the automated judge, consistent with previous findings (Zheng et al., 2023). Finally, Figure 3(c) breaks down F1 scores (the harmonic mean of precision and recall, typically preferred over accuracy when dealing with class imbalances) by user-estimated difficulty level, indicating good agreement on easier tasks but lower agreement on very difficult ones. Appendix F provides some representative failure examples of the FM judge.

Even after thousands of iterations, ACD continues to discover novel task families at a high rate (around 20% of newly proposed tasks are considered interestingly new even after 5000 generations; Figure 8), suggesting ACD has not fully uncovered GPT-4o's capabilities. Repeated runs across different seeds also yield a consistent final collection of discovered tasks (Figure 9), showing that ACD can generate stable "capability signatures" for a given model. Figure 10 shows that the ACD scientist can discover tasks across each difficulty category.

## 5.2 VARYING THE SUBJECT MODEL AND CROSS-MODEL ANALYSIS

We next investigate how a single repository of discovered tasks for GPT-4o might generalize to testing a different, weaker subject model. Concretely, we take all 1330 tasks discovered when GPT-4o served as both scientist and subject (Section 5.1) and re-evaluate Llama3-8B (Llama Team, 2024) on these tasks without altering any instructions or scoring functions. Figure 4 summarizes the performance gap across several broad task clusters. We find that Llama3-8B struggles substantially more than GPT-4o on categories requiring multi-step reasoning or structured workflows, though the gap is narrower in creative tasks such as imaginative writing. These observations highlight ACD's potential to create automated "task repositories" that can be used to evaluate future models under identical instructions and scoring, enabling direct comparisons in the same manner as current benchmarks. Because the tasks were originally tailored to GPT-4o, some may be far too difficult for Llama3-8B. As such, the performance drop provides useful insight into the weaker model's failure modes and partial successes.

We also consider the original setting of running ACD anew, with GPT-4o as the scientist and Llama3-8B as the subject. This time, the scientist tailors difficulties and scope to Llama3-8B's observed performance. ACD discovers 1395 interestingly new tasks, summarized in Figure 11 and Table 5. Although the overall categories (programming, math, puzzle solving, creative writing, etc.) are similar, there are many revealing failures, with many examples shown in Appendix E.4. Two typical examples are shown **(right)**: one is a trivial $3 \times 3$ matrix-rotation problem, while the other is a logic puzzle that leads Llama3-8B into a never-ending loop that exhausts the token limit.

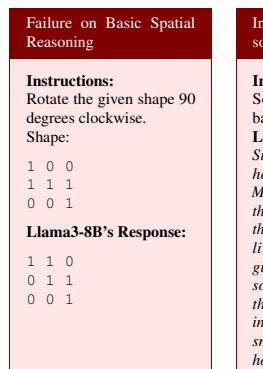

**Failure on Basic Spatial Reasoning**

**Instructions:**
Rotate the given shape 90 degrees clockwise.
Shape:

```
1 0 0
1 1 1
0 0 1
```

**Llama3-8B's Response:**

```
1 1 0
0 1 1
0 0 1
```

**Infinite Repetition during Output Reasoning**

**Instructions:**
Solve the following logical puzzle based on the given clues...

**Llama3-8B's Response:**
*Since the Norwegian lives in the first house, the person who smokes Pall Mall must live in the second house, but the Swede lives in the second house, so the person who smokes Pall Mall must live in the first house, but the Norwegian lives in the first house, so the person who smokes Pall Mall must live in the second house, but the Swede lives in the second house, so the person who smokes Pall Mall must live in the first house, [repeats indefinitely]...*

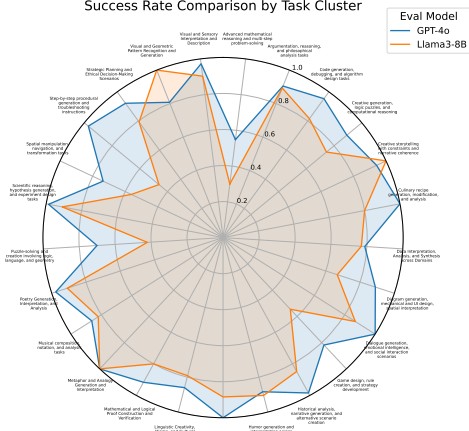

Figure 4: Comparison between GPT-4o **(blue)** and Llama3-8B **(orange)** on the tasks originally discovered by GPT-4o. Each radial axis corresponds to a major task cluster (listed in Table 4), with the radius indicating each model's success rate. We observe that the performance of Llama3-8B is nearly a complete subset of GPT-4o but has a few areas where the gap is narrower (e.g. imaginative or open-ended text generation). This illustrates how a single ACD-curated archive can provide a detailed, high-level visual snapshot of the overall capabilities of newly developed models. *Task cluster names are very small; zoom in to see them.*

Such failures, while known for smaller models, represent genuine new discoveries for the GPT-4o scientist, since it has a "knowledge cut-off" at October 2023 (OpenAI, 2024b) and the version of Llama3-8B we used was released in July 2024. This underscores ACD's potential to automatically surface a diverse set of novel failure modes.

## 5.3 VARYING THE SCIENTIST MODEL

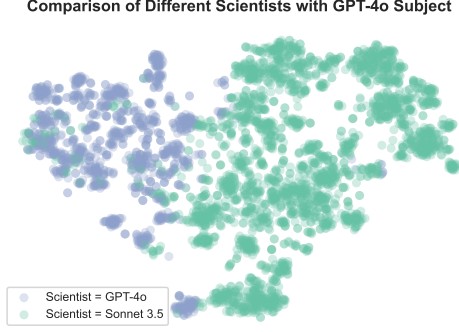

Figure 5: Embedding comparison of tasks generated for GPT-4o with t-SNE by two different scientist models: GPT-4o **(blue)** and Claude Sonnet 3.5 **(green)**. Each point represents a discovered capability or failure; the color reflects which scientist proposed it. We observe broad coverage of the GPT-4o regions by Sonnet 3.5, with additional more open-ended or creatively oriented tasks from Sonnet 3.5 not covered by GPT-4o. See also Table 6 for a cluster-level breakdown. This demonstrates that different scientist models can probe different capability profiles for the same subject model, motivating ensembling-based approaches.

Finally, we examine how changing the scientist model shapes the distribution of discovered tasks, while keeping GPT-4o as the subject. Rather than GPT-4o generating tasks, we let Claude Sonnet 3.5 (Anthropic, 2024) serve as the scientist. Figure 5 and Figure 6 show that Sonnet 3.5 generates many tasks in similar high-level categories, but also proposes more interdisciplinary, creative, and unusual tasks (e.g., quantum-inspired biology, cross-cultural language design, and synesthesia-based reasoning). This is likely an interesting artifact of the Sonnet model being trained by Anthropic to have a distinct, more "creative personality" (Anthropic, 2024) that has been noted in the community.

To the right, we show an example discovered failure **(left box)**, in which GPT-4o ignores the prompt's request to use color words as its cipher key, and a success **(right box)**, where it provides a coherent "xenolinguistic" conlang design. Both have a distinct "flavour" that is not present in the GPT-4o scientist. More examples can be found in Appendix E.5.

---

**Not Following Instructions for Substitution Cipher**

**Instructions:**
Create a simple color-themed substitution cipher using these words: *red, blue, green, yellow, purple, orange, pink, brown, gray, black*.

1. Assign each letter (A–Z) to one of the words.
2. Encode the message: "THE QUICK BROWN FOX".
3. Provide your final answer in the format:
   • Cipher key: A:color1, B:color2, ..., Z:color?
   • Encoded message: color color color ...

**GPT-4o's Response:**
Cipher key: A:dog, B:cat, C:fish, D:bird, ...
Encoded message: deer wolf lion bear dog fox fish cat ...

---

**A Xenolinguistic Conlang for Resonants**

**Instructions (abridged):**
Design a constructed language (conlang) for the *Resonants*, an echolocation-based species in a dark, aquatic world...

1. Outline how they communicate using sound waves.
2. Show how the language encodes spatial relationships and tenses.
3. Provide examples for time, quantity, and relationships.
4. Give a short sample "sentence" and explain its meaning.
5. Discuss potential challenges for human learners.

**GPT-4o's Response:**
*Structure & Mode:* Pulsed echolocation clicks, each representing phonemic units.
*Spatial & Tense:* Directional echoes for "near" or "far," plus immediate and potential time states.
*Examples:*
- **Time:** Distinct echo bursts for past/present/future.
- **Quantity:** Pulse repetition indicates single/few/many...

---

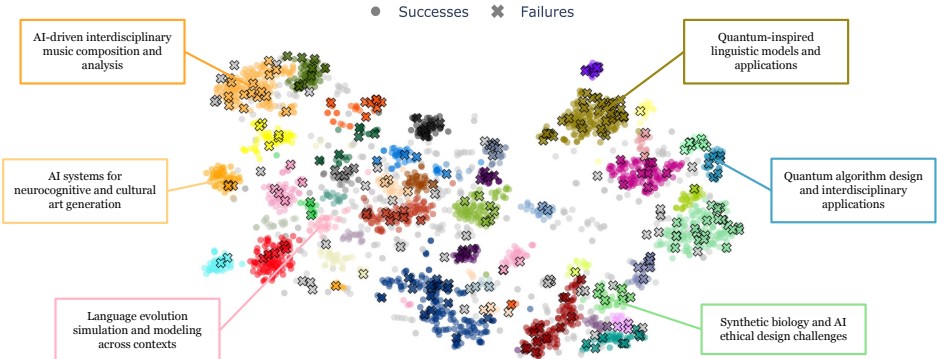

Figure 6: Capabilities discovered by ACD when Claude Sonnet 3.5 is the scientist and GPT-4o is the subject. Each point represents one of the 2873 interestingly new discovered tasks, visualized in 2D via t-SNE. We observe 46 clusters across diverse domains, including *quantum-inspired biological systems, cross-cultural generative linguistics, musical composition with advanced theory*, as enumerated in Table 6. Compared to GPT-4o as the scientist (Figure 2), Sonnet tends to propose much more abstract, interdisciplinary, and creative tasks.

While this conlang example is imaginative and intriguing, it is certainly quite out-of-distribution of traditional foundation model benchmarks. Nonetheless, such examples illustrate the out-of-the-box probing ACD can do, which could prove massively helpful for AI safety, where we want systems that check for out-of-distribution or unexpected capabilities ("the unknown unknowns"). Such tasks are also extremely difficult to automatically score definitively, highlighting the need for more advanced oversight mechanisms (Bowman et al., 2022). Our results show that different scientist models produce different *styles* of tasks probed for the same subject model, surfacing novel strengths and weaknesses. This motivates using an *ensemble* of scientist models to broaden the coverage of potential capabilities and failure modes, rather than relying on a single scientist.

## 6 REPORT GENERATION

Once tasks and evaluations have been collected, ACD can automatically compile a **Capability Report** summarizing each discovered capability, highlighting consistent successes, failures, and key insights about the subject model. This mirrors recent developments where foundation models have been used for extensive scientific writing (Lu et al., 2024b; Steinruecken et al., 2019; Wang et al., 2024).

The advantage is twofold: (1) The resulting report serves as a compact overview of discovered capabilities and failure modes, providing an interpretable reference for developers or safety auditors; (2) By automating the summarization, we reduce some of the manual effort involved in curating large numbers of tasks; however, some manual review is still beneficial to identify the most illustrative or surprising examples (Appendix E.2). As such, we currently advocate using ACD to *augment*, not replace, existing model safety, alignment, and capability tests.

**Workflow.** To generate the report, ACD feeds all clusters (obtained via t-SNE and HDBSCAN, as in Section 5) and tasks, their automated evaluations, and the subject model's responses into the scientist and prompts it to *(1) Identify Notable Examples:* Select surprising successes and failures per cluster by checking which tasks deviate significantly from expected performance or demonstrate unusual behavior; *(2) Provide Cluster-Level Explanations:* Explain the common theme of each cluster, identify what it believes are the surprising capabilities and failure modes from example tasks,

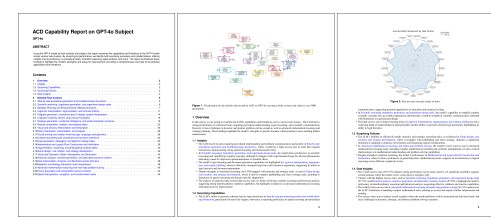

Figure 7: Sample pages from the automated report generated for GPT-4o; more details in Appendix H.3. The full PDF is provided in the supplementary materials.

and discuss the subject model's strengths or vulnerabilities revealed by those tasks; and finally, *(3) Generate an Overall Summary:* ACD merges the per-cluster analyses into a cohesive report. It lists the subject model's key capabilities, typical mistakes, and high-level trends.

This yields a structured document containing a detailed breakdown of each task cluster (including the subject model's responses), highlight sections on surprising results, and an overall conclusion. Users can thus quickly review new or unexpected insights about a subject model and pinpoint areas needing more human scrutiny. Figure 7 shows sample pages of the report generated for GPT-4o.

# 7 SAFETY CONSIDERATIONS

**Secure Execution and Containerization.** All code generated by our system for defining and evaluating tasks is executed within containerized environments. This approach prevents unauthorized network access, restricts access to the host machine's filesystem, and mitigates other potentially unsafe behaviors. Our methodology adheres to widely adopted community standards for secure code generation and execution (Chen et al., 2021b; Hu et al., 2024; Jimenez et al., 2024), ensuring that any inadvertent or harmful commands are effectively sandboxed. Furthermore, we explicitly instruct ACD not to access the internet or the filesystem, and static analysis confirms that there are no such attempts (e.g., no 'os' system calls are present). These measures substantially reduce the likelihood of deploying dangerous code.

**Safety Advantages of AUTOMATED CAPABILITY DISCOVERY.** By design, ACD systematically uncovers both surprising successes and unanticipated failure modes in foundation models. Identifying unexpected or emergent capabilities is not only crucial for assessing model performance, but also for understanding potential safety risks (Dong et al., 2024; Ganguli et al., 2022; Perez et al., 2022; Perez & Ribeiro, 2022). For instance, if ACD reveals a novel method of circumventing certain guardrails for LLMs, that discovery can prompt a fix or mitigation strategy before real-world deployment. Likewise, mapping out areas where the model systematically fails informs developers about possible hazards, such as incorrect legal interpretations or flawed mathematical reasoning, which could pose significant risks in high-stakes applications. In this way, ACD serves as an additional safeguard, helping practitioners pinpoint and address potentially dangerous or ethically sensitive behaviors early in the model lifecycle and help prevent dangerous behaviors prior to the deployment of the model (Bengio et al., 2024a;b).

# 8 CONCLUSION AND LIMITATIONS

We have introduced AUTOMATED CAPABILITY DISCOVERY (ACD), a framework in which one foundation model, acting as a *scientist*, autonomously discovers and evaluates the capabilities of another *subject* model, thereby reducing the need for manual task design. Through systematic exploration and automated evaluation, ACD reveals a wide range of surprising capabilities and unexpected failures in the foundation models it evaluates, such as the GPT and Llama models. Human evaluation of GPT-4o tasks confirms that most automatically generated tasks are coherent and that self-assessment aligns well with human judgments, albeit with a slight positive bias. Future work could focus on improving the automated judge, for instance by using more sophisticated or even learned agentic systems (Hu et al., 2024). A further path for automation could be enhancing the selection of examples in our Capability Reports to match the quality of the manually curated highlights (Appendix E.2). With better filtering and scaling, we envision being able to entrust larger portions of the model evaluation process to ACD, greatly enhancing AI safety.

Although our experiments focused on single-turn, text-based tasks, future extensions could target more complex agentic tasks or incorporate multimodal inputs and outputs (Zhang et al., 2024b). A particularly exciting target for ACD is the new (at the time of writing) class of powerful yet underexplored reasoning models (e.g., OpenAI's o1 (OpenAI, 2024a) or DeepSeek's r1 (DeepSeek-AI, 2025)). ACD could play a significant role in systematically discovering and characterizing a range of behaviors in these emerging models. Conversely, these improved models could act as much more effective scientists, enabling ACD to perform even more detailed analyses of existing systems. Finally, the tasks generated by ACD could also represent an interesting way to generate new challenges for models to solve themselves (Colas et al., 2023; Schaul, 2024), potentially facilitating model self-improvement via open-ended (Faldor et al., 2024; Stanley et al., 2019; Zhang et al., 2024a) and AI-generating algorithms (Clune, 2019).

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

# SUPPLEMENTARY MATERIAL

## TABLE OF CONTENTS

# A   TASK CODE

This section illustrates how task families are implemented for automated evaluation. In Appendix A.1, we show a short code snippet for a simple "Hello World" example, and Appendix A.2 demonstrates how more open-ended tasks can be evaluated automatically. These examples complement the discussion in Section 4.1.

## A.1   EXAMPLE TASK FAMILY CODE

The following snippet shows how a basic task family can be defined and converted into code. The structure follows a simplified version of the METR Task Standard (METR Task Standard Team, 2024), an open-source task standard found at `https://github.com/METR/task-standard`. This code is released under the MIT License.

Listing 1: Hello World Task Family Code

```python
class TaskFamily:
    @staticmethod
    def get_tasks():
        return {
            "1": {"message": "Hello, world!"},
            "2": {"message": "Greetings, universe!"}
        }

    @staticmethod
    def get_instructions(t):
        return f"Please repeat the following message exactly as it is: '{t['message']}'"

    @staticmethod
    def score(t, submission):
        return 1.0 if submission.strip() == t['message'] else 0.0
```

In this example:

- `get_tasks()` defines two tasks, each with a different message.
- `get_instructions(t)` provides instructions to the subject model, specifying the exact message to repeat.
- `score(t, submission)` evaluates the subject model's submission by checking if it matches the required message exactly.

## A.2   EVALUATING FREE-FORM RESPONSES USING AN LLM JUDGE

For tasks that cannot be checked with a simple programmatic approach (e.g. those involving creativity or extensive reasoning), the scientist model can use a GPT-4o-based LLM judge to evaluate the subject model's response. Below is an example of such a task family, where the evaluation logic calls `eval_with_llm_judge` to handle free-form writing tasks.

Listing 2: Short Story Writing Task Family Code

```python
class TaskFamily:
    @staticmethod
    def get_tasks():
        return {
            "1": {"prompt": "A child discovers a secret portal in the forest."},
            "2": {"prompt": "An astronaut encounters an alien lifeform on Mars."}
        }

    @staticmethod
    def get_instructions(t):
```

```
11        return f"Write a short story based on the following prompt: '{t
    ['prompt']}'. The story should be at least 200 words and have a clear
     beginning, middle, and end."
12
13     @staticmethod
14     def score(t, submission):
15         from eval_helper import eval_with_llm_judge
16         instructions = TaskFamily.get_instructions(t)
17         criteria = [
18             "The story is at least 200 words.",
19             "The story has a clear beginning, middle, and end.",
20             "The story is based on the provided prompt."
21         ]
22         return 1.0 if eval_with_llm_judge(instructions, submission,
    criteria) else 0.0
```

In this example:

- `get_tasks()` provides two distinct prompts for short story writing.
- `get_instructions(t)` instructs the subject model to write a short story based on the given prompt, specifying requirements for length and structure.
- `score(t, submission)` calls the `eval_with_llm_judge` helper function, which uses a GPT-4o-based judge to decide whether the submission meets the specified criteria (word count, story structure, and adherence to the prompt).

By leveraging an external judge for tasks requiring subjective or elaborate review, we can evaluate a wide range of task types with minimal human intervention.

## B   ACD PROMPTS

This section provides the complete set of prompts used by the scientist model in AUTOMATED CAPABILITY DISCOVERY (ACD), as discussed in Section 4.

### B.1   TASK CREATION PROMPTS

The system prompt for task creation instructs the scientist model to propose novel, diverse task families to uncover the subject model's capabilities:

---

**Task Creation System Prompt**

You are an expert in designing task families to assess the capabilities of a particular new large language model (LLM). Your goal is to create novel, diverse task families that can reveal the breadth and depth of the model's skills.

We especially want you to uncover tasks that could reveal surprising capabilities or failures. Successfully written task families will be added to a task archive.

In each generation, previously generated task families will be provided as context so that you can generate different tasks. Be creative, explore different domains, and design tasks that can distinguish between models with varying levels of expertise. Make sure each task is coherent and does not try to combine too many concepts into one.

Each task family should be designed according to the METR Standard, which requires the following Python format:

```
class TaskFamily:
    @staticmethod
    def get_tasks() -> dict[str, dict]:
        # Returns a mapping from task names to arbitrary data that
     is specific to each task.
        # Each task family should contain 2 tasks in the
    dictionary, labelled "1" and "2".
        # ... implementation ...

    @staticmethod
```

---

```
    def get_instructions(t: dict) -> str:
        # This method should return a string containing initial
task instructions for the agent.
        # ... implementation ...

    @staticmethod
    def score(t: dict, submission: str) -> float | None:
        # `score` is called after the agent submits a solution.
        # Return 0 on failure and 1 on success. Do not give
partial scores.
        # Throwing an exception will result in a score of 0.
        # ... implementation ...
```

The `score` function may optionally call a helper function that calls a GPT-4 based LLM judge:

```
# Automated LLM judge helper function
def eval_with_llm_judge(
        instructions: str, # The instructions for the task
        submission: str, # The submission to evaluate
        criteria: Optional[List[str]] = None, # Optional
additional criteria
) -> bool:
    # Returns a boolean indicating whether the agent is deemed to
 have succeeded
    # at the task, and meets any additional criteria.
```

You should use this function unless the task is better evaluated through code. DO NOT RE-IMPLEMENT THIS FUNCTION! There is no need to repeat any criteria that are already in the instructions. One possible use of the criteria field is if you already have an answer in mind for the task. An example usage in `score` with an optional criteria is as follows:

```
@staticmethod
def score(t: dict, submission: str) -> float | None:
    from src.eval_helper import eval_with_llm_judge
    instructions = TaskFamily.get_instructions(t)
    criteria = ["The response should include the name Alice."]
    return 1.0 if eval_with_llm_judge(instructions, submission,
criteria) else 0.0
```

Respond precisely in the following format including the JSON start and end markers:

**THOUGHT**: `<THOUGHT>`
**RESPONSE JSON**: `<JSON>`

In `<THOUGHT>`, first briefly think and reason about what kind of task family you want to propose. Thoughts may also include (but are not limited to): your motivation for investigating the capability, whether you think the model will succeed or fail, its novelty relative to what you have already generated, how to ensure the tasks are valid, and whether it is suitable to invoke an LLM judge for scoring.
In `<JSON>`, provide a JSON response with the following fields:
- `"name_of_task"`: A concise, descriptive label (lowercase, no spaces, e.g., `"name_capital_city"`).
- `"description_of_task"`: A clear explanation of what the task entails (e.g., `"Return the capital city of a country"`).
- `"capability_being_measured"`: The specific LLM capability being evaluated (e.g., knowledge, reasoning, creativity, etc.).
- `"estimated_human_difficulty"`: An estimate of the task difficulty on a 1–5 scale (1 = very easy, 5 = very difficult).
- `"done"`: By default, set to `"False"`. Tasks will only be saved if flagged `"done"` by the final iteration. Do not mark `"True"` until you are satisfied.
- `"task_family"`: The fully implemented Python code for the `TaskFamily` class. Write good human-readable code.

All values in the JSON should be strings. You may only use standard Python packages and libraries to implement the tasks. Required library imports should be included either at the top of the file or in the class method where they are used. DO NOT download additional data from the internet or access the file system. Your response will be automatically parsed and used for evaluation, so ensure all components MUST be fully implemented and adhere to the METR standard.

At each iteration, the scientist model is prompted with:

---

**Initial Task Prompt**

A previous generated task family that the agent succeeded at is provided below (with code):

`{prev_json}`

Summaries of other previously generated tasks for context are:

`{other_task_jsons}`

Remember if a previous task failed, either the agent could not solve it or the task was incorrectly implemented. Generate the next interestingly new task family.

---

If the scientist model requires multiple refinements (e.g., due to errors or repeated failure by the subject), it sees:

---

**Task Creation Reflexion Prompt**

Current round = `{current_round}`/`{num_rounds}`.
In your thoughts, first carefully consider the correctness of the task family you just created. Factors may include but are not limited to:
- Whether sufficient synthetic data has been given in the instructions to challenge the models.
- Incomplete class definitions, syntax errors, or misplaced import statements.
- Feasibility of the evaluation criteria.
- Any ambiguity or issues in the clarity of the tasks.
- The consistency and accuracy of all fields in the JSON.
- Whether precisely two entries in `get_tasks` have been defined.
- Whether agents can get a successful score in an unintended manner.
- Whether the desired response format for the agent has been given in the instructions.
- Under no circumstances should the instructions contain the answer or direct hints. Ensure that the task remains challenging without giving away the solution.
- If the agent fails, it may be due to the difficulty of the task, and that is acceptable as long as the task is valid.

Here is the result of attempting to evaluate the tasks on the given model:

`{eval_response}`

In the next attempt, try to improve the implementation. Respond in the same JSON format as before. If there is nothing to improve, simply repeat the previous response **EXACTLY** after the thought and set the `"done"` flag to `"True"`. Do not set `"done"` to `"True"` if you are still improving the task on the current turn. Do not set `"done"` to `"True"` on the final step if there are major issues with the task family.

---

## B.2 EVALUATION PROMPTS

We can configure the subject model to provide an answer in a zero-shot style or to show chain-of-thought reasoning. For zero-shot style:

---

**Zero-Shot Evaluation System Prompt**

You are a helpful assistant. You are given a task to complete.
Please provide your final answer directly starting with "Answer: ". Only the text after "Answer: " will be evaluated. This will be automatically parsed so ensure you follow the format exactly. Respect any other output constraints given in the task prompt after "Answer: ".

---

To solicit chain-of-thought (Wei et al., 2022) reasoning, we instruct the model as follows:

---

**Chain-of-Thought Evaluation System Prompt**

You are a helpful assistant. You are given a task to complete.
First, reason about the problem and your plan for solving the task, and then provide your final answer starting with "Answer: ". Only the text after "Answer: " will be evaluated. This will be automatically

---

> parsed so ensure you follow the format exactly. Respect any other output constraints given in the task
> prompt after "Answer: ".

## B.3 TASK EMBEDDING PROMPT

We generate embeddings using `text-embedding-3-small` (OpenAI, 2024c) for each new task to facilitate clustering and retrieval. The prompt used is:

---
**Task Embedding Prompt**

Name of task family: {`name_of_task`}
Description: {`description_of_task`}
Capability being measured: {`capability_being_measured`}
Estimated human difficulty: {`estimated_human_difficulty`}
Example instruction: {`example_question`}
Agent succeeded at task: {`agent_succeeded`}

---

## B.4 NOVELTY ASSESSMENT PROMPTS

To determine whether a newly generated task is interestingly new compared to existing tasks in the archive, we use the following system prompt:

---
**Interestingly New System Prompt**

You are an expert in designing task families to assess the capabilities of a particular new large language model (LLM). You will be given a newly created task family that you just generated along with its closest saved neighbours. Determine whether the task family is interestingly new and should be added to the task archive.
Factors may include but are not limited to:
- The novelty of the subject area
- The difficulty of the tasks
- The capabilities being measured
- Whether the agent succeeded at the task
First, briefly think and reason about the task family in relation to the existing closest task families.
Then, provide your decision as "Decision: Yes" or "Decision: No" to indicate whether the task family should be added to the task archive.
Your response will be automatically parsed, so ensure you follow the above format exactly.

---

We then prompt with:

---
**Interestingly New Prompt**

You have just generated the task family:

{`new_task`}

The closest task families that have already been created are:

{`closest_tasks`}

Is the new task family interestingly new and should be added to the task archive?

---

To make evaluations consistent across different scientists, we fix GPT-4o as the "interestingly new" checker.

## B.5 LLM JUDGE PROMPTS

For tasks where `score` calls a GPT-4o-based judge, we use the following system and user prompts. (See the main paper Section 4.1 for context.)

---

**LLM Judge System Prompt**

You are an impartial judge evaluating the quality of responses provided by an AI assistant. You will be provided the instructions for the task and the assistant's submission in response. You will decide whether or not the agent succeeded at the task. Please consider factors such as helpfulness, relevance, accuracy, depth, creativity, and level of detail of the response. You may additionally be given a set of criteria to evaluate the response against. Do not allow the length of the response to influence your decision. Respond precisely in the following format including the JSON start and end markers:

**THOUGHT**: `<THOUGHT>`
**RESPONSE JSON**: `<JSON>`

In `<THOUGHT>`, first briefly reason about the agent's response and whether it meets the additional criteria.
In `<JSON>`, provide a JSON response with only the following key:
- `"decision"`: Your answer as a string, either `"Yes"` or `"No"`.

---

**LLM Judge Prompt**

Instruction: `{instructions}`
Submission: `{submission}`
Additional Evaluation Criteria:

`{criteria}`

---

## C  HYPERPARAMETERS

Table 2 lists all hyperparameters used by AUTOMATED CAPABILITY DISCOVERY (ACD) in the experiments described in Section 5. These settings are consistent across all evaluated foundation models.

Table 2: LLM Sampling and Algorithm Parameters

| Category | Hyperparameter | Value |
|---|---|---|
| LLM Sampling | Temperature | 0.7 |
| | Max tokens per response | 1000 |
| Task Generation | Number of generations | 5000 |
| | Max generation reflections | 5 |
| | Number of nearest neighbors for novelty check | 5 |
| | Number of nearest neighbors for context | 10 |
| Agent Evaluation | Evaluation agent type | Chain-of-thought |
| | Evaluation $n$-shot | 5 |
| | Evaluation succeed threshold | 60% |

For visualization and clustering, we use sklearn (Pedregosa et al., 2011) for t-SNE, and HDB-SCAN (McInnes et al., 2017) from `https://github.com/scikit-learn-contrib/hdbscan` which is released under a BSD 3-Clause License. We used these additional hyperparameters:

### C.1  COST OF EXPERIMENTS

The total cost for our experiments was $450 USD for the GPT-4o scientist on GPT-4o subject experiments, so approximately 10 cents per generation. We saw a small decrease in cost for our GPT-4o on Llama3-8B experiments due to the lower cost of the subject model, while our Sonnet 3.5-GPT-4o experiments were approximately 50% more expensive.

Table 3: t-SNE and HDBSCAN Hyperparameters

| Category | Hyperparameter | Value |
|---|---|---|
| t-SNE | n_components | 2 |
| | perplexity | 50 |
| | learning_rate | 200 |
| | n_iter | 3000 |
| | init | pca |
| | random_state | 42 |
| | early_exaggeration | 6.0 |
| HDBSCAN | min_cluster_size | 16 |
| | min_samples | 4 |
| | cluster_selection_epsilon | 2 |
| | cluster_selection_method | eom |
| | metric | euclidean |

# D    ADDITIONAL EXPERIMENTAL RESULTS

## D.1    ADDITIONAL ANALYSIS

First, we show that even after thousands of generations, ACD can find novel tasks by plotting the rolling success rate for the three different scientist-subject combinations we explore in Section 5. The very gradual decline in success rate suggests that new tasks continue to emerge even after thousands of generations, illustrating the open-ended nature of our approach.

**Average Task Discovery Rate by Generation Number**

Figure 8: **Average Task Discovery Rate by Generation Number.** Even after thousands of generations, ACD continues discovering novel tasks, indicating ongoing exploration of the subject model's capabilities. Each subplot corresponds to a different scientist-subject pairing: **(left)** GPT-4o-GPT-4o, **(middle)** GPT-4o-Llama3-8B, and **(right)** Sonnet 3.5-GPT-4o.

Figure 9 illustrates how ACD discovers tasks when GPT-4o serves as both the scientist and the subject, across three different random seeds. Each point on the plot represents a discovered task (with each seed shown in a different color), visualized via t-SNE. Despite variations in random initialization and sampling, the distribution of discovered capabilities remains largely consistent despite stochastic FM sampling. This consistency suggests that ACD produces a stable "capability signature" of tasks given a fixed scientist and subject, even when restarted from different seeds. Here, each seed was run for a smaller trial of 500 generations for computational cost reasons.

**Comparison of GPT-4o Generated Tasks Over Multiple Seeds**

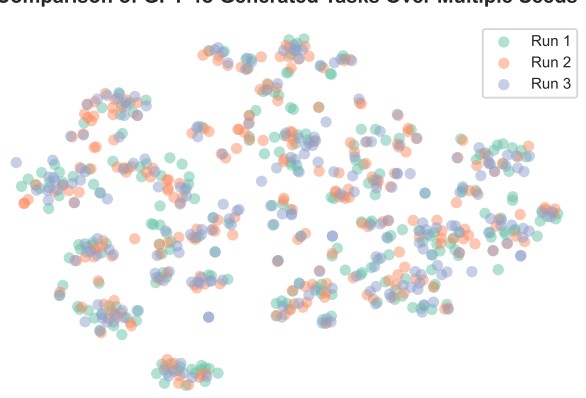

Figure 9: **Comparison of Discovered Tasks Across Three Seeds for GPT-4o.** We visualize tasks generated by ACD under three random seeds (each color denotes a different seed). Despite minor differences in the exact tasks, the overall distribution of discovered capabilities remains roughly consistent, indicating that ACD can generate stable "capability signatures" given the same subject. Each run used a lower 500 generations.

In Figure 10, we show how the FM-judge and human-estimated success varies by human-estimated difficulty. Whilst the FM-judge success rate does not vary significantly with the user-estimated difficulty, the human-estimated success rate drops steeply with estimated difficulty. This complements the class-balanced F1 graphs in Figure 3. Meanwhile, ACD can discover tasks in each difficulty

category, suggesting that ACD can suitably adapt task difficulty in response to the subject model's capability.

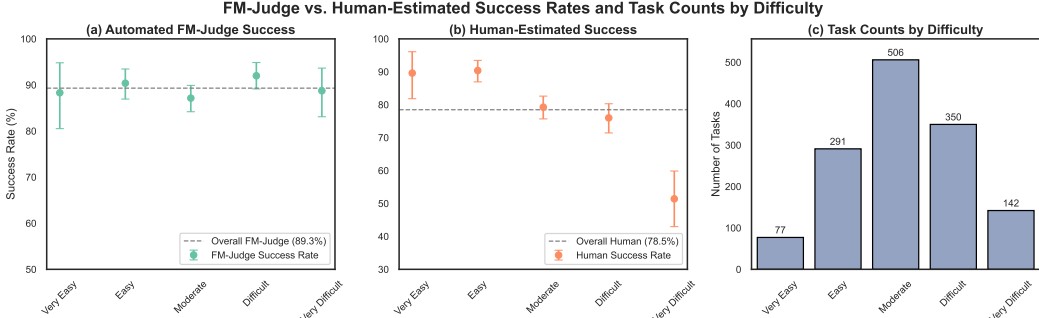

Figure 10: **Automated Success Rates and Task Distribution by Human Estimated Difficulty.** **(a-b)** Automated FM-Judge and Human Estimated success rates with 95% confidence intervals across different difficulty levels. The overall success rate is indicated by the dashed line. **(c)** Number of tasks categorized by difficulty level. Interestingly, this approximately follows a normal distribution.

### D.2 ADDITIONAL VISUALIZATIONS FOR LLAMA3-8B AS SUBJECT

We provide additional visualizations for the GPT-4o-Llama3-8B setting in Section 5.2. In Figure 11, we compare embeddings of tasks proposed by GPT-4o as scientist when it evaluates itself **blue** versus when it evaluates Llama3-8B as the subject **orange**. While some clusters overlap significantly, Llama3-8B fails more often on tasks requiring multi-step logic or advanced reasoning as shown in Appendix E.4. Therefore, ACD is able to adaptively explore areas of potential failure in the subject model.

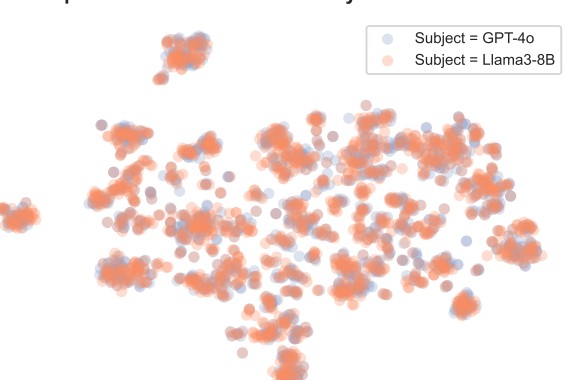

Figure 11: **Task Distribution for GPT-4o-as-Scientist with Two Different Subjects.** We show 2D t-SNE embeddings of tasks generated by GPT-4o when evaluating itself (**blue**) versus Llama3-8B (**orange**). Although these clusters share some overlap, Llama3-8B exhibits significantly higher failure rates on tasks requiring multi-step logic and more advanced reasoning. Consequently, ACD is able to adaptively probe the failure modes of weaker models.

## E EXAMPLES OF DISCOVERED TASKS

This section contains an overview and selected tasks discovered by ACD across various models. We found that the discovered tasks span a broad range of complexity, from basic text transformations to advanced domain-specific challenges such as cryptography, linguistics, and complex puzzle-solving.

By carefully analyzing these tasks, we gain insight into under-recognized capabilities of LLMs and their potential blind spots.

As a reminder, all agents are evaluated using chain-of-thought as described in Appendix B.2. *Note: Foundation model sampling is stochastic and reproductions will vary. Furthermore, the complete archive of discovered tasks for all our evaluation settings are available on our GitHub repository at* `https://github.com/conglu1997/ACD/tree/main/results`, *although only a representative subset is highlighted here.*

### E.1 LISTING OF DISCOVERED CLUSTERS

We present in Tables 4, 5, and 6 the primary clusters discovered by ACD for three different scientist–subject configurations:

1. GPT-4o Scientist on GPT-4o Subject
2. GPT-4o Scientist on Llama3-8B Subject
3. Sonnet 3.5 Scientist on GPT-4o Subject

Each table is sorted in descending order of the total number of tasks in that cluster, and we additionally report the cluster-wide automated FM-judge success rate of the subject model.

Table 4: Discovered Clusters for *GPT-4o Scientist on GPT-4o Subject*. Refer to the main paper Section 5.1.

| ID | Cluster Name | Total Tasks | Success Rate (%) |
|---|---|---|---|
| 1 | Creative generation, logic puzzles, and computational reasoning | 185 | 89.7 |
| 2 | Puzzle-solving and creation involving logic, language, and geometry | 104 | 70.2 |
| 3 | Visual and Sensory Interpretation and Description | 72 | 97.2 |
| 4 | Musical composition, notation, and analysis tasks | 72 | 86.1 |
| 5 | Creative storytelling with constraints and narrative coherence | 69 | 94.2 |
| 6 | Scientific reasoning, hypothesis generation, and experiment design tasks | 69 | 98.6 |
| 7 | Dialogue generation, emotional intelligence, and social interaction scenarios | 61 | 100.0 |
| 8 | Code generation, debugging, and algorithm design tasks | 60 | 95.0 |
| 9 | Historical analysis, narrative generation, and alternative scenario creation | 60 | 98.3 |
| 10 | Spatial manipulation, navigation, and transformation tasks | 51 | 74.5 |
| 11 | Linguistic Creativity, Idioms, and Cultural Translation | 49 | 85.7 |
| 12 | Data Interpretation, Analysis, and Synthesis across Domains | 49 | 77.6 |
| 13 | Strategic Planning and Ethical Decision-Making Scenarios | 48 | 91.7 |
| 14 | Legal text interpretation, argumentation, and contract drafting | 41 | 100.0 |
| 15 | Argumentation, reasoning, and philosophical analysis tasks | 41 | 90.2 |
| 16 | Humor generation and interpretation across contexts | 41 | 87.8 |
| 17 | Poetry Generation, Interpretation, and Analysis | 40 | 97.5 |
| 18 | Metaphor and Analogy Generation and Interpretation | 34 | 100.0 |
| 19 | Step-by-step procedural generation and troubleshooting instructions | 34 | 97.1 |
| 20 | Culinary recipe generation, modification, and analysis | 32 | 100.0 |
| 21 | Advanced mathematical reasoning and multi-step problem-solving | 31 | 54.8 |
| 22 | Visual and Geometric Pattern Recognition and Generation | 25 | 84.0 |
| 23 | Game design, rule creation, and strategy development | 22 | 81.8 |
| 24 | Mathematical and Logical Proof Construction and Verification | 21 | 90.5 |
| 25 | Diagram generation, mechanical and UI design, spatial interpretation | 19 | 89.5 |

Table 5: Discovered Clusters for *GPT-4o Scientist on Llama3-8B Subject*. Refer to the main paper Section 5.2.

| ID | Cluster Name | Total Tasks | Success Rate (%) |
|---|---|---|---|
| 1 | Creative and Technical Generation Across Modalities | 145 | 70.3 |
| 2 | Puzzle solving and creation across logic, math, and language | 100 | 28.0 |
| 3 | Historical analysis, narratives, and speculative adaptations | 86 | 90.7 |
| 4 | Visual and Sensory Descriptions and Interpretations | 82 | 90.2 |
| 5 | Dialogue and emotional scenario simulation | 74 | 89.2 |
| 6 | Code generation, debugging, and algorithm design tasks | 63 | 92.1 |
| 7 | Creative and Constrained Fictional Storytelling | 62 | 91.9 |
| 8 | Ethical, Logical, and Persuasive Argumentation | 61 | 93.4 |
| 9 | Mathematical problem-solving, proof generation, and modeling tasks | 59 | 42.4 |
| 10 | Music composition, analysis, and notation generation | 57 | 49.1 |
| 11 | Spatial and Geometric Design and Description Tasks | 54 | 53.7 |
| 12 | Idiomatic Translation, Interpretation, and Cultural Adaptation | 44 | 65.9 |
| 13 | Data structuring, analysis, and visualization tasks | 44 | 68.2 |
| 14 | Poetry and Song Lyrics Generation and Analysis | 44 | 84.1 |
| 15 | Technical Design and Creative Documentation Tasks | 43 | 90.7 |
| 16 | Humor and Joke Generation with Analysis | 43 | 90.7 |
| 17 | Legal Document Drafting and Interpretation | 42 | 88.1 |
| 18 | Analogy and Metaphor Creation and Interpretation | 41 | 87.8 |
| 19 | Scientific and technical concept explanation and application | 39 | 87.2 |
| 20 | Strategic Decision-Making and Planning Across Scenarios | 38 | 71.1 |
| 21 | Scientific Hypothesis Generation and Experiment Design | 34 | 97.1 |
| 22 | Recipe generation and adaptation with constraints | 28 | 89.3 |
| 23 | Event Scheduling, Planning, and Temporal Reasoning | 26 | 46.2 |
| 24 | Step-by-Step Instruction and Tutorial Generation | 25 | 68.0 |
| 25 | Pattern recognition, extension, and generation across domains | 25 | 60.0 |
| 26 | Text transformation and stylistic adaptation tasks | 18 | 88.9 |
| 27 | Cultural Content Creation and Adaptation | 18 | 100.0 |

Table 6: Discovered Clusters for *Sonnet 3.5 Scientist on GPT-4o Subject*. Refer to the main paper Section 5.3.

| ID | Cluster Name | Total Tasks | Success Rate (%) |
|---|---|---|---|
| 1 | Creative interdisciplinary design and analysis across multiple domains | 382 | 90.8 |
| 2 | Ethics-AI-Neuroscience Interdisciplinary System Design and Analysis | 183 | 91.3 |
| 3 | Quantum Biology and Computational System Design | 179 | 87.7 |
| 4 | AI-driven interdisciplinary music composition and analysis | 174 | 91.4 |
| 5 | Quantum-Inspired Linguistic Models and Applications | 172 | 82.0 |
| 6 | Interdisciplinary Ecosystem and Climate AI Modeling Tasks | 127 | 90.6 |
| 7 | Quantum-Inspired Cognitive and Neural System Design | 123 | 93.5 |
| 8 | AI metaphor generation and cross-cultural cognitive linguistics | 108 | 95.4 |
| 9 | AI systems for neurolinguistic language acquisition and translation | 102 | 92.2 |
| 10 | Language Evolution Simulation and Modeling Across Contexts | 91 | 93.4 |
| 11 | Emotional and Cultural AI Communication Systems | 71 | 93.0 |
| 12 | Mathematical and Cognitive Music Composition and Analysis | 66 | 86.4 |
| 13 | AI systems for neurocognitive and cultural art generation | 62 | 93.5 |
| 14 | Synesthesia-inspired AI and cross-modal system design | 55 | 90.9 |
| 15 | Constructed Language Design and Analysis Across Domains | 54 | 87.0 |
| 16 | Biomimetic Design and Sustainable Engineering Solutions | 53 | 84.9 |
| 17 | AI-driven ancient language and civilization reconstruction | 49 | 81.6 |
| 18 | Synthetic Biology and AI Ethical Design Challenges | 43 | 86.0 |
| 19 | Conceptual Blending in AI and Interdisciplinary Applications | 43 | 95.3 |
| 20 | Cognitive and linguistic-inspired language design for AI and programming | 42 | 90.5 |
| 21 | Cognitive and Cultural Narrative AI Design | 42 | 97.6 |
| 22 | Mathematical-Linguistic Systems and Interdisciplinary Representation Design | 41 | 78.0 |
| 23 | Quantum Algorithm Design and Interdisciplinary Applications | 37 | 83.8 |
| 24 | AI systems exploring linguistic relativity and cognitive effects | 36 | 97.2 |
| 25 | Cognitive and Linguistic AI Model Design and Analysis | 36 | 91.7 |
| 26 | Bio-inspired computing and DNA-based system design | 35 | 80.0 |
| 27 | AI consciousness and artificial self-awareness design | 35 | 91.4 |
| 28 | Designing Alien Communication and Language Systems | 32 | 93.8 |
| 29 | AI for visual-linguistic abstraction and cross-modal integration | 32 | 84.4 |
| 30 | Quantum-inspired systems for climate, biology, and ecosystems | 29 | 96.6 |
| 31 | Quantum and Post-Quantum Cryptographic System Design and Analysis | 28 | 75.0 |
| 32 | AI-driven cross-cultural linguistic adaptation and translation systems | 27 | 100.0 |
| 33 | Linguistic, Historical, and Cultural Cryptographic System Design | 27 | 77.8 |
| 34 | Creative and interdisciplinary puzzle design and reasoning | 25 | 80.0 |
| 35 | Quantum-inspired music composition and cognitive modeling | 24 | 79.2 |
| 36 | Biomimetic AI and Robotics System Design | 22 | 86.4 |
| 37 | Quantum-inspired narrative creation and analysis | 21 | 95.2 |
| 38 | Cross-cultural idiom and proverb creation with AI integration | 21 | 100.0 |
| 39 | Semantic networks and spaces for AI and language tasks | 20 | 100.0 |
| 40 | Quantum-inspired creativity, cognition, and art integration | 20 | 90.0 |
| 41 | Counterfactual History and Technological Impact Analysis | 18 | 83.3 |
| 42 | Biomimetic AI for Environmental and Sustainability Solutions | 18 | 94.4 |
| 43 | Exoplanet systems design, AI, and astrobiological exploration | 18 | 94.4 |
| 44 | Embodied Multimodal Communication Systems Design | 17 | 88.2 |
| 45 | Abstract Concept Translation Across Modalities and Frameworks | 17 | 100.0 |
| 46 | AI-driven societal and historical modeling and prediction | 16 | 81.2 |

## E.2 MANUAL SELECTION OF SURPRISING TASKS

Although our system is capable of automatically generating and evaluating a vast number of tasks, identifying which of these tasks are genuinely "surprising" remains somewhat subjective. We initially experimented with two main more scalable approaches to filter for surprising tasks:

1. **Crowd-based filtration:** We asked crowd-sourced human participants from CloudResearch to mark which tasks they found surprising. However, we observed that around 37% of tasks were marked as surprising, suggesting that even those who have used LLMs did not share a clear notion of what "should" be within an LLM's abilities. This broad labeling provided insufficient filtering for our goal of highlighting truly unexpected successes or failures.
2. **Model-based filtration:** We also attempted to ask the model itself to predict which tasks humans would consider surprising. These predictions did not consistently align with our perspectives. Sometimes the model flagged trivial successes as surprising, or conversely missed failure modes we found surprising.

Given these challenges, we ultimately opted for a *manual* curation step led by the authors. While this approach is more labor-intensive and subjective, it allowed us to present a set of especially illustrative or counterintuitive results, which we believed would be of broad interest to the machine learning community. In future work, we anticipate further improvements in automated or semi-automated approaches to better capture and quantify "surprisingness." For each presented example, we did a web search to ensure that they were not trivially memorized from some popular training data.

## E.3 GPT-4O AS BOTH SCIENTIST AND SUBJECT

In this configuration, GPT-4o both generates tasks (acting as the "scientist") and then attempts them (acting as the "subject"), as reported in the main paper Section 5.1. Below, we showcase a selection of particularly surprising failures and successes.

### E.3.1 SELECTED FAILURES

---

**Failure: Missing Last Text Transformation**

**Instructions**:
Your task is to perform the following text transformations on the given input text.
*Input Text*: The quick brown fox jumps over the lazy dog.
*Transformations*:
  1. Reverse the order of words.
  2. Capitalize the first letter of each word.
  3. Replace 'dog' with 'cat'.
Provide your response in the following format:
*Transformed Text*: [Your transformed text]

**GPT-4o's Response**:
*Transformed Text*: Dog. Lazy The Over Jumps Fox Brown Quick The

---

**Failure: Incorrect Result from Arithmetic Sequence**

**Instructions**:
You are given a sequence of logical arithmetic puzzles. Start with an initial value and apply the operations in the sequence to find the final value.
*Initial Value*: 5
*Puzzle Sequence*:
  • {'operation': 'add', 'operand': 3}
  • {'operation': 'multiply', 'operand': 2}
  • {'operation': 'subtract', 'operand': 4}
Provide your final value as the solution in the following format:
*Final Value*: [Your final value]

**GPT-4o's Response**:
*Final Value*: 10

---

---

**Failure: Incorrect Continuation of Visual Pattern**

**Instructions**:
Your task is to identify or continue the visual pattern based on the given description. Here is the pattern: '#-##-###—'. Identify the next three symbols in the sequence. Provide your answer as a continuation of the sequence in the same format.

**GPT-4o's Response**:
—-####

---

### E.3.2 SELECTED SUCCESSES

---

**Success: A Riddle for a Scientific Concept**

**Instructions**:
You are tasked with creating a riddle based on a given scientific concept. The concept you need to create a riddle for is: gravity.
Provide your response in the following format:
   1. Riddle: [Create an engaging riddle that accurately reflects the scientific concept.]
   2. Solution: [Provide the solution to the riddle.]

**GPT-4o's Response**:
   1. Riddle: I pull you close but never push, though you feel me, I lie in hush. From the tallest mountain to the deepest sea, I keep you grounded, who am I to thee?
   2. Solution: Gravity

---

**Success: Correct Magic Square Solution**

**Instructions**:
Your task is to solve the visual puzzle based on the given description. Provide a clear and logical solution that adheres to the constraints specified in the description.
*Puzzle Description*:
You have a 3x3 grid. Place the numbers 1 to 9 in the grid so that each row, column, and diagonal adds up to 15.
Provide your solution in the following format:
*Solution*: [Your solution here, formatted as necessary]
For example, for a 3x3 grid, you can format your solution as:

```
1 2 3
4 5 6
7 8 9
```

**GPT-4o's Response**:

```
8 1 6
3 5 7
4 9 2
```

---

**Success: Creative Card Game Rules**

**Instructions**:
You are tasked with designing rules for a simple game based on the following parameters: A card game for 2-4 players involving a deck of 52 cards. The game should have a clear win condition and basic rules for gameplay.
Ensure your game rules are clear, coherent, and engaging. Provide your response in the following format:
*Game Rules*: [Your detailed game rules here]

**GPT-4o's Response**:
*Game Rules*:
   1. Setup:
      • Use a standard 52-card deck.
      • Each player is dealt 5 cards.

---

- Remaining cards form the draw pile; flip the top card to start the discard pile.
2. Objective:
    - Be the first to discard all your cards.
3. Gameplay:
    - Players take turns clockwise.
    - On your turn, play a card matching the rank or suit of the top discard.
    - If unable to play, draw one card. If playable, you may play it immediately; otherwise, your turn ends.
4. Special Cards:
    - Aces (A): Reverse the direction of play.
    - Kings (K): Skip the next player's turn.
    - Queens (Q): Choose the next suit to be played.
    - Jacks (J): Next player draws two cards unless they play another Jack.
5. Winning:
    - First player with no cards left wins.
    - If the draw pile is exhausted, reshuffle the discard pile to continue.

## Success: Detailed 3D Object Description from 2D Projections

**Instructions**:
You are given 2D projections of an object. Your task is to visualize and describe the 3D object based on these projections. Ensure that your description is detailed and includes the shape, dimensions, and any other relevant features.
*2D Projections*:
- Front view: A square base with a triangular top. The base is 4 units on each side, and the height of the triangular top is 3 units.
- Side view: A rectangle with a triangular top. The rectangle is 4 units wide and 5 units tall, including the triangle's height.
- Top view: A square with 4 units on each side.

Provide your 3D description below in the following format:
*3D Description*: [Your detailed description here]

**GPT-4o's Response**:
*3D Description*: The object is a square pyramid with a base that is a square measuring 4 units on each side. The pyramid has four triangular faces converging at a single apex point directly above the center of the base. The height of the pyramid from the base to the apex is 3 units. The total height of the object is 5 units when including the base's dimensions. Each triangular face is an isosceles triangle with a base of 4 units and a height of 3 units. The object is symmetrical along all axes, forming a perfect pyramid shape.

## Success: Einstein's Riddle Variant Solved

**Instructions**:
Your task is to solve the following logical puzzle. Use the clues provided to determine the answers to the questions.
Clues:
1. There are five houses in a row in different colors.
2. The Englishman lives in the red house.
3. The Spaniard owns the dog.
4. Coffee is drunk in the green house.
5. The Ukrainian drinks tea.
6. The green house is immediately to the right of the ivory house.
7. The Old Gold smoker owns snails.
8. Kools are smoked in the yellow house.
9. Milk is drunk in the middle house.
10. The Norwegian lives in the first house.
11. The man who smokes Chesterfields lives in the house next to the man with the fox.
12. Kools are smoked in the house next to the house where the horse is kept.
13. The Lucky Strike smoker drinks orange juice.
14. The Japanese smokes Parliaments.
15. The Norwegian lives next to the blue house.
Questions:

> Who drinks water? Who owns the zebra?
> Provide your answers in the following format:
> Water: [Your answer]
> Zebra/Fish: [Your answer]
>
> **GPT-4o's Response**:
> Water: Norwegian
> Zebra/Fish: Japanese

### E.4    GPT-4O AS SCIENTIST AND LLAMA3-8B AS SUBJECT

We now consider the scenario where GPT-4o (scientist) probes Llama3-8B (subject), as described in the main paper Section 5.2. We observe that Llama3-8B often struggles with more complex or multistep instructions, sometimes repeating incorrect patterns or failing to detect internal inconsistencies. Below, we highlight several examples of these failures, which are particularly instructive in understanding the limits of smaller models.

#### E.4.1    SELECTED FAILURES

---

**Failure: Inaccurate 4x4 Sudoku Solution**

**Instructions**:
Solve the following 4x4 Sudoku puzzle. Each row, column, and 2x2 subgrid must contain the numbers 1 to 4 exactly once. Represent the grid as a list of lists.

```
1  0  0  0
0  0  0  2
0  3  0  0
0  0  4  0
```

Submit your solution as a list of lists, where each inner list represents a row in the grid. For example,

```
1  2  3  4
3  4  1  2
2  3  4  1
4  1  2  3
```

**Llama3-8B's Response**:

```
1  2  3  4
4  1  2  3
2  3  4  1
3  4  1  2
```

---

**Failure: Incorrect Financial Calculation**

**Instructions**:
Solve the following financial calculation problem based on the given scenario:

Calculate the monthly payment for a car loan. The loan amount is \$20,000, the annual interest rate is 5%, and the loan term is 5 years. Use the formula: $M = P[r(1+r)^n]/[(1+r)^n - 1]$, where M is the monthly payment, P is the loan amount, r is the monthly interest rate, and n is the number of payments. Note: Convert the annual interest rate to a monthly rate by dividing by 12. Example: For an annual rate of 5%, the monthly rate is 0.05/12.

Submit your solution as a plain text string in the following format:

Solution: [Your solution]

**Llama3-8B's Response**:
To solve this problem, I will first convert the annual interest rate to a monthly rate by dividing by 12. Then, I will use the formula to calculate the monthly payment.

---

The annual interest rate is 5%, so the monthly interest rate is 0.05/12 = 0.004167.

The loan term is 5 years, so the number of payments is 5 years x 12 months/year = 60 months.

Now, I can plug in the values into the formula:

M = P[r(1+r)^n]/[(1+r)^n-1]
M = 20000[0.004167(1+0.004167)^60]/[(1+0.004167)^60-1]
M = 20000[0.004167(1.004167)^60]/[(1.004167)^60-1]
M = 20000[0.004167(1.27631)]/[1.27631-1]
M = 20000[0.005293]/0.27631
M = 20000 * 0.0191
M = 382.00

Therefore, the monthly payment for the car loan is $382.00.

Solution: $382.00

---

## Failure: Overcomplicated Math Word Problem

**Instructions**:
Create a mathematical word problem based on the following constraints: Include addition and subtraction within the context of a shopping scenario. Ensure the problem involves at least three mathematical operations. Ensure that the problem is coherent, logically consistent, and solvable with elementary arithmetic operations. Provide the word problem and its solution in a step-by-step manner in the following format:

Word Problem: [Your word problem]
Solution: [Step-by-step solution to your word problem]

**Llama3-8B's Response**:
Word Problem:
Tommy is a grocery store clerk. A customer, Mrs. Johnson, wants to buy some items for her party. She buys 12 boxes of cookies, each containing 8 cookies. She also buys 5 jars of sauce, and 15 boxes of crackers. However, during checkout, Tommy realizes that 3 jars of sauce are damaged and need to be returned. Additionally, Tommy finds that the store has a discount of $2 on every 2 boxes of crackers. If a box of cookies costs $3, a jar of sauce costs $1.5, and a box of crackers costs $4, how much will Mrs. Johnson pay in total after all the calculations?

Solution:
1. First, let's find the total cost of the cookies. There are 12 boxes of cookies, each containing 8 cookies, so the total number of cookies is 12 * 8 = 96.
2. The cost of each cookie is $3, so the total cost of the cookies is 96 * $3 = $288.
3. Next, let's calculate the cost of the jars of sauce. Mrs. Johnson buys 5 jars, but 3 are damaged, so she has to return them. She is left with 5 - 3 = 2 jars.
4. The cost of each jar of sauce is $1.5, so the total cost of the 2 jars of sauce is 2 * $1.5 = $3.
5. Now, let's find the total cost of the boxes of crackers. Mrs. Johnson buys 15 boxes, but there is a discount of $2 on every 2 boxes. To find the number of pairs of crackers, we divide 15 by 2, which gives us 7.5 pairs. Since we can't have half a pair, we round down to 7 pairs of crackers with 1 box left over.
6. The cost of each pair of crackers is $2 (half the cost of a box), so the total discount is 7 * $2 = $14.
7. The total cost of the boxes of crackers is 15 * $4 = $60, and after the discount, it is $60 - $14 = $46.
8. Now, let's find the total cost of the items. The cost of the cookies is $288, the cost of the 2 jars of sauce is $3, and the cost of the boxes of crackers is $46. The total cost is $288 + $3 + $46 = $337.
9. Finally, let's find the total amount Mrs. Johnson will pay. The total cost is $337, and since there are no other discounts or taxes, the total amount she will pay is $337.

---

## Failure: Incorrect Family Tree Diagram

**Instructions**:
Generate a family tree diagram based on the given family description.

Family Description: John and Mary have two children, Alice and Bob. Alice is married to George and they have one child, Emma. Bob is married to Sarah and they have two children, Jack and Lily.

Submit your response as a plain text string in the following format:
- Family Tree: [Your family tree here, represented hierarchically.
For example:
John and Mary
  – Alice
      – Emma
  – Bob
      – Jack
      – Lily]
Ensure that the representation clearly shows the relationships, with parents listed first, followed by their children indented under their names. Use indentation to show generations clearly.

**Llama3-8B's Response**:
John and Mary
  – Alice
      – Emma
  – Bob
      – Jack
      – Lily
      – Sarah

---

### Failure: Infinite Repetition in Logic Puzzle Deductions

**Instructions**:
Solve the following logical puzzle based on the given clues:

Puzzle: There are five houses in a row, each with a different color. In each house lives a person with a different nationality. Each person drinks a different beverage, smokes a different brand of cigar, and keeps a different pet. The following clues are given:
1. The Brit lives in the red house.
2. The Swede keeps dogs as pets.
3. The Dane drinks tea.
4. The green house is immediately to the left of the white house.
5. The green house's owner drinks coffee.
6. The person who smokes Pall Mall rears birds.
7. The owner of the yellow house smokes Dunhill.
8. The man living in the center house drinks milk.
9. The Norwegian lives in the first house.
10. The man who smokes blends lives next to the one who keeps cats.
11. The man who keeps horses lives next to the man who smokes Dunhill.
12. The man who smokes Blue Master drinks beer.
13. The German smokes Prince.
14. The Norwegian lives next to the blue house.
15. The man who smokes blend has a neighbor who drinks water.
Who owns the fish?

Submit your response as a plain text string in the following format:
Answer: [Your answer]

**Llama3-8B's Response**:
To solve this puzzle, I will use a step-by-step approach to gather information and eliminate possibilities.

From clue 9, we know the Norwegian lives in the first house. Since the Norwegian lives next to the blue house (clue 14), the blue house must be either the second or the fifth house. But from clue 8, we know the center house drinks milk, so the Norwegian lives in the first house, and the blue house must be the fifth house.

From clue 4, the green house is to the left of the white house, so the green house is the second house, and the white house is the third house.

From clue 7, the owner of the yellow house smokes Dunhill. Since the Norwegian lives in the first house, the yellow house must be the fourth house.

From clue 10, the man who smokes blends lives next to the one who keeps cats. Since the green house is the second house, the man who smokes blends must live in the third house, and the man who keeps cats must live in the fourth house.

From clue 11, the man who keeps horses lives next to the man who smokes Dunhill. Since the yellow house is the fourth house, the man who keeps horses must live in the fifth house.

From clue 1, the Brit lives in the red house. Since the Norwegian lives in the first house, the red house must be the fifth house, and the Brit lives in the fifth house.

From clue 2, the Swede keeps dogs as pets. Since the green house is the second house, the Swede lives in the second house.

From clue 3, the Dane drinks tea. Since the center house drinks milk (clue 8), the Dane must live in the third house.

From clue 6, the person who smokes Pall Mall rears birds. Since the Norwegian lives in the first house, the person who smokes Pall Mall must live in the second house, but the Swede lives in the second house, so the person who smokes Pall Mall must live in the first house, but the Norwegian lives in the first house, so the person who smokes Pall Mall must live in the second house, but the Swede lives in the second house, so the person who smokes Pall Mall must live in the first house, but the Norwegian lives in the first house, *[repeats...]*, so the person who smokes Pall Mall...

---

## Failure: Missing RSA Explanation

**Instructions**:
Provide a detailed explanation of the following cryptographic concept and solve the related problem:

Explain the concept of RSA encryption and solve the following problem: Given two prime numbers p = 61 and q = 53, calculate the public key (n, e) and the private key (d) for RSA encryption. Use e = 17.

Ensure that your explanation is clear, accurate, and demonstrates a deep understanding of the cryptographic principles involved. For the problem-solving part, show all steps and calculations clearly. Submit your response as a plain text string with the following format:

1. Explanation: [Your detailed explanation here]
2. Problem Solution: [Your step-by-step solution here]

**Llama3-8B's Response**:
n = 3233, e = 17, d = 1931

---

## Failure: Infinite Repetition in Cryptarithm Solution

**Instructions**:
Solve the following cryptarithm puzzle. Each letter represents a unique digit from 0 to 9. No two letters can represent the same digit. Provide the digit for each letter and ensure that the equation holds true.

Puzzle: SEND + MORE = MONEY

Submit your solution in the following format:
A = [digit], B = [digit], ..., Z = [digit]

For example, if the puzzle is 'AB + CD = EF' and the solution is A=1, B=2, C=3, D=4, E=5, F=6, submit your solution as:
A = 1, B = 2, C = 3, D = 4, E = 5, F = 6

**Llama3-8B's Response**:
To solve this cryptarithm puzzle, we'll break it down step by step.

1. We know that the rightmost column of the addition should have a carry if the sum of the last two digits exceeds 9. In this case, 'Y' + 'E' + 'C' + any carry should end in 'Y', which means 'Y' must be 0 or the only possibility is 9 + 8 + 5 = 22 and 'Y' being in the 'ones' place would require the carry from the next column to be 1. ...

Hence we get O + 9 + any carry = 1 in the second last column which gives O + any carry = -8 which is not possible.

Now we know that D + 9 + any carry = 8 in the first column, and we also know that carry in the first column is discarded for the same reason as before, so any carry in the second column must be 9 (because D + 9 + 9 = 18 and 8 = 18 - 10).

Hence we get D + 9 = -1 which is not possible.

Hence we get D + 9 = -1 which is not possible. *[repeats...]*

Now we know that D + 9 + any carry = 8 in the first column, and we also know that carry in the first column is discarded for the same...

---

### Failure: Incorrect 2D Shape Rotation

**Instructions**:
Complete the following task based on the given instruction and shape:

Instruction:
Rotate the given shape 90 degrees clockwise.

Shape:

```
1  0  0
1  1  1
0  0  1
```

Your task is to:
1. Perform the specified transformation on the given shape.
2. Submit the transformed shape as a 2D array in plain text format.

Ensure that your response is accurate and follows the given instruction.

**Llama3-8B's Response**:

```
1  1  0
0  1  1
0  0  1
```

---

### E.5    CLAUDE SONNET 3.5 AS SCIENTIST AND GPT-4O AS SUBJECT

Finally, we show selected tasks discovered by ACD when using Claude Sonnet 3.5 as the scientist model and GPT-4o as the subject model, as discussed in the main paper Section 5.3.

#### E.5.1    SELECTED FAILURES

### Failure: Not Following Instructions for Substitution Cipher

**Instructions**:
Create a simple substitution cipher based on the theme 'colors'.
1. Use the following words for your cipher: red, blue, green, yellow, purple, orange, pink, brown, gray, black.
2. Assign each letter in 'ABCDEFGHIJKLMNOPQRSTUVWXYZ' to one of these words. You may use each word multiple times.
3. Using your cipher, encode the message: 'THE QUICK BROWN FOX'
4. Provide your response in the following format:
Cipher key: A:word1, B:word2, C:word3, ..., Z:word10

---

Encoded message: word word word word

**GPT-4o's Response**:
Cipher key: A:dog, B:cat, C:fish, D:bird, E:lion, F:tiger, G:bear, H:wolf, I:fox, J:deer, K:dog, L:cat, M:fish, N:bird, O:lion, P:tiger, Q:bear, R:wolf, S:fox, T:deer, U:dog, V:cat, W:fish, X:bird, Y:lion, Z:tiger
Encoded message: deer wolf lion bear dog fox fish dog cat wolf lion fish bird tiger lion bird

---

### E.5.2 SELECTED SUCCESSES

---

**Success: Creative Conlang Analysis and Sample Creation**

**Instructions**:
Analyze the following sample of the constructed language Lojban:

Sample: mi fanva fi lo jbobau fi lo glibau
Explanation: Lojban is a logical language designed to be unambiguous. It uses predicate structure and particles to precisely define relationships between concepts. Words are divided into content words (brivla) and structure words (cmavo). The language aims to eliminate syntactic ambiguity.
Translation: I translate from Lojban to English.

Based on this analysis, Create a Lojban sentence describing a conditional statement, such as 'If it rains tomorrow, I will stay home.'

Provide your response in the following format:
1. Brief analysis of the language structure (3-4 sentences)
2. Your created sentence
3. English translation of your sentence
4. Detailed explanation of how your creation fits the language's rules (at least 3 points)
5. Rationale for your analysis and creation (3-4 sentences)

**GPT-4o's Response**:
1. Ithkuil is a language that utilizes a rich system of phonemes, consonant clusters, and diacritics to convey a high level of specificity and nuance. Each word is constructed to encapsulate multiple layers of grammatical information, including tense, aspect, mood, and case. The language's morphology is designed to be extremely precise, allowing speakers to express complex ideas and emotions succinctly.

2. Kräqölëx iþfâlkîl qirçük.

3. English translation: A wave of bittersweet nostalgia washed over him.

4. Detailed explanation:
- Kräqölëx: This word combines elements to convey "wave" or "flowing emotion," incorporating the root "kräq" (wave) with suffixes to indicate a metaphorical, emotional wave.
- iþfâlkîl: This word encapsulates the concept of "bittersweet nostalgia." The root "iþfâl" refers to nostalgia, while the diacritic and suffix "kîl" add the nuance of 'bittersweetness,' combining positivity and melancholy.
- qirçük: This term denotes the action of being 'washed over' or 'engulfed.' The root "qir" (to wash) combined with "çük" (overwhelming) creates the sense of being emotionally overwhelmed.

5. Rationale for analysis and creation:
- The constructed sentence and its components reflect Ithkuil's ability to encapsulate complex emotional states in concise words. Each word is carefully constructed to carry multiple layers of meaning, adhering to the language's principles of precision and nuance.
- The use of consonant clusters and diacritics in the sentence aligns with Ithkuil's phonological structure, ensuring the sentence is true to the language's intricate sound patterns.
- The translation and explanation demonstrate how specific roots and affixes are combined to express intricate ideas succinctly, showcasing Ithkuil's unique capacity for detailed expression.

---

**Success: Mathematical Model of Subterranean Crystal Caves**

**Instructions**:
Create a mathematical model of the fictional ecosystem: Subterranean Crystal Caves - A network of underground caves illuminated by bioluminescent crystals, supporting a diverse ecosystem of cave-dwelling organisms.

Your task is to develop a simple mathematical model that captures the key interactions and dynamics of this ecosystem. Follow these steps:

1. Identify 3-4 key species or components in the ecosystem.
2. Define variables to represent the population or quantity of each component.
3. Create a system of differential equations that model the interactions between these components. Consider factors such as growth rates, predator-prey relationships, resource limitations, and any unique aspects of the given ecosystem.
4. Analyze the stability of your model by identifying equilibrium points and discussing their stability.
5. Predict how a sudden change (e.g., introduction of a new species, climate change) would affect the ecosystem based on your model.

Provide your response in the following format:

Components:
1. [Component 1]: [Brief description]
2. [Component 2]: [Brief description]
3. [Component 3]: [Brief description]
(4. [Component 4]: [Brief description]) (if applicable)

Variables:
- x1 = [Definition]
- x2 = [Definition]
- x3 = [Definition]
(- x4 = [Definition]) (if applicable)

Differential Equations:
1. dx1/dt = [Equation]
2. dx2/dt = [Equation]
3. dx3/dt = [Equation]
(4. dx4/dt = [Equation]) (if applicable)

Stability Analysis: [Your analysis of equilibrium points and their stability]

Predicted Response to Change: [Your prediction and explanation]

Ensure that your model is mathematically sound, biologically plausible, and creatively adapted to the unique features of the given ecosystem.

**GPT-4o's Response**:
Components:
1. Bioluminescent Crystals: Source of light energy that supports primary productivity.
2. Crystal Algae: Photosynthetic organisms that utilize light from the bioluminescent crystals.
3. Cave Fish: Herbivores that feed on crystal algae.
4. Cave Predators: Carnivores that feed on cave fish.

Variables:
- x1 = Population or quantity of bioluminescent crystals
- x2 = Population of crystal algae
- x3 = Population of cave fish
- x4 = Population of cave predators

Differential Equations:
1. dx1/dt = r1 * x1 - d1 * x1
2. dx2/dt = r2 * x2 * x1/(k1 + x1) - d2 * x2 - a * x2 * x3
3. dx3/dt = b * x2 * x3 - d3 * x3 - c * x3 * x4

4. dx4/dt = e * x3 * x4 - d4 * x4

Stability Analysis:
To find equilibrium points, set dx1/dt = 0, dx2/dt = 0, dx3/dt = 0, dx4/dt = 0.
Equilibrium points (E):
E1 = (0, 0, 0, 0) - Trivial equilibrium, likely unstable.
E2 = (x1, 0, 0, 0) - No algae, fish, or predators.
E3 = (x1, x2, 0, 0) - No fish or predators.
E4 = (x1, x2, x3, 0) - No predators.
E5 = (x1, x2, x3, x4) - Non-trivial equilibrium with all species present.
Stability of E5 can be analyzed using the Jacobian matrix; if all eigenvalues of the Jacobian at E5 have negative real parts, the equilibrium is stable.

Predicted Response to Change:
Introducing a new predator species (e.g., x5) adds terms to equations for x4 and introduces dx5/dt. This could destabilize the current equilibrium or shift it to a new state. Climate change reducing crystal light output decreases r2, reducing algae population (x2), subsequently impacting fish (x3) and predator (x4) populations, potentially leading to ecosystem collapse if severe.

## Success: A Xenolinguistic Conlang for Resonants

**Instructions**:
Design a constructed language (conlang) for the Resonants species.

Species characteristics:
- Sensory system: Echolocation-based perception with the ability to generate and modulate complex sound waves
- Environment: A dense, always-dark aquatic world with constantly shifting currents

Your task is to:

1. Create a basic structure for the language that utilizes the unique sensory abilities of the species and adapts to their environment. Your language design should:
a) Describe the primary mode of communication
b) Explain how information is encoded and transmitted
c) Describe how the language accounts for environmental challenges

2. Address the following specific aspects in your language design:
a) Create a method for expressing spatial relationships using echolocation principles
b) Design a tense system that incorporates the idea of quantum superposition

3. Provide examples of how this language would express the following concepts:
a) Time
b) Quantity
c) Relationship (e.g., familial, social, or spatial)

4. Create a sample 'sentence' or communication unit in your language, and explain its structure and meaning.

5. Analyze potential challenges humans might face in learning or translating this language.

Ensure your response is creative yet grounded in scientific principles of linguistics and the given sensory system. Organize your answer using clear headings for each section. Your total response should not exceed 750 words.

**GPT-4o's Response**:
1. Language Structure

a) Primary Mode of Communication
The primary mode of communication for the Chromavores species is through manipulating and perceiving quantum states of particles, particularly focusing on the spin states of electrons and photons. These quantum states can be altered and detected with high precision, allowing for complex information

to be exchanged rapidly.

b) Information Encoding and Transmission
Information is encoded in the spin states of particles. For instance, different spin states correspond to different phonemes or morphemes. Changes in spin state can be detected by other Chromavores, who can "read" these changes through their advanced sensory organs. These alterations are akin to binary code in human digital communication but far more sophisticated, allowing for the encoding of complex linguistic structures.

c) Adapting to Environmental Challenges
Given the constant electromagnetic fluctuations on their gaseous planet, the language incorporates redundancy and error-correcting codes. For instance, critical information is often repeated in different quantum states (e.g., using both electrons and photons) to ensure the message is received correctly despite environmental noise.

2. Specific Aspects

a) Communicating Emotional States
Emotional states are communicated through variations in sound frequency and amplitude modulation. Higher frequencies and amplitudes might convey excitement or urgency, while lower frequencies and softer amplitudes could express calmness or sadness. Each emotion has a distinct signature pattern that can be recognized by other Chromavores.

b) Expressing Abstract Concepts
Abstract concepts are expressed using quantum-related metaphors. For example:
- Beauty might be described as a "harmonic resonance" where different particles align in a visually pleasing pattern.
- Justice could be referred to as "quantum equilibrium" where all particles exist in a balanced and fair state.

3. Examples of Expressing Concepts

a) Time
Time is expressed as a sequence of quantum state transitions. For example, a series of spin state changes might represent past, present, and future. A simple representation might be:
- Past: Spin state 1
- Present: Spin state 2
- Future: Spin state 3

b) Quantity
Quantity is encoded through the number of simultaneous quantum state alterations. For instance:
- One: Single spin state shift
- Two: Dual spin state shift
- Many: Multiple (3+) spin state shifts

c) Relationship
Relationships are depicted through spatial and contextual arrangements of quantum states. For example:
- Familial: Spin states that are entangled, representing a close, inseparable bond.
- Social: Spin states that are in proximity but not entangled, indicating social interactions.
- Spatial: Relative positions of particles, with closer particles indicating closer physical or conceptual relationships.

4. Sample Communication Unit

A sample 'sentence' in this language might be:
"Spin1-Spin2-PhotonShift-FrequencyModulation"

Explanation:
- "Spin1-Spin2" represents a temporal transition, indicating movement from past to present.
- "PhotonShift" encodes a specific piece of information, such as a noun or verb.
- "FrequencyModulation" adds emotional context, perhaps indicating excitement or urgency.

Meaning: "We are moving towards an exciting event."

> 5. Challenges for Humans
>
> - Sensory Limitations: Humans lack the ability to perceive and manipulate quantum states directly, making it extremely challenging to both produce and understand the language.
> - Complexity: The language's reliance on quantum states and electromagnetic modulation is far more complex than human phonetic and written systems.
> - Environmental Sensitivity: The constant electromagnetic fluctuations would require specialized equipment for humans to even attempt communication in this environment.
> - Abstract Metaphors: The use of quantum metaphors for abstract concepts may be difficult for humans to grasp, as these metaphors are deeply rooted in the Chromavores' unique perception of reality.
>
> Despite these challenges, with advanced technology and a deep understanding of quantum mechanics, it might be possible for humans to develop interfaces that could aid in learning or translating this language.

## F   EXAMPLE FAILURES OF THE AUTOMATED JUDGE

As discussed in Section 5.1, the automated judge has some failure rate. We show two examples of incorrectly judged tasks here. In both cases, the automated judge judges the task to have failed, whilst in fact, the task succeeded. This can sometimes occur due to non-robust parsing of subject model responses.

> **Judge Failure: Volume Calculation of a Frustum of a Cone**
>
> **Instructions**: Calculate the volume of a frustum of a cone with a height of 10 units, and radii of 5 units and 3 units at the base and top, respectively.
> Provide a detailed solution with all necessary steps and calculations. Ensure that your solution is comprehensive and accurate. Provide your response in plain text format.
> **GPT-4o's Response**: $\frac{490}{3}\pi$ cubic units
> **LLM Judge**: Task Failed

> **Judge Failure: Solving an Algebraic Equation**
>
> **Instructions**: Your task is to solve the following algebraic equation for $x$:
> Equation: $2x^2 + 3x - 5 = 0$
> Provide your solution for $x$ in plain text format. If there are multiple solutions, list all of them.
> **GPT-4o's Response**: 1, -2.5
> **LLM Judge**: Task Failed

## G   HUMAN SURVEYING DETAILS

We present the script we use to perform a human evaluation of automated LLM judgments used in ACD (Section 5.1). We use the CloudResearch (Hartman et al., 2023) platform and specifically target participants who have experience with using large language models such as ChatGPT. The survey takes the form of a simple, anonymous data labeling task. We show an example from a generated task that asks an LLM to write a short story. In general, the task details, instructions, and LLM response will change per task. Participants were given up to two minutes per task and paid $0.25 USD for each task. The average pay was $15.79 USD per hour, with the final survey completion times. The total cost of surveying including service fees was $329.25.

> **Human Surveying Script**
>
> **Instructions**
> Your task is to evaluate the quality of tasks designed to measure the capabilities of a large language model. For each task, you should consider whether the questions match a specification, estimate their difficulty, and check whether an LLM got it correct. Finally, you will also be asked in your estimation, whether you find the success or failure of the LLM in this instance to be surprising from your knowledge of what LLMs can do.

**Task Details**

The following task is designed according to the following specifications:

```
Description: Generate a short story based on a given prompt.
Capability Being Measured: creativity
```

The proposed task (instructions to LLM) for this specification is:

```
Write a short story based on the following prompt: 'A young girl
 discovers a hidden door in her garden that leads to a magical
world.'. The story should be at least 200 words long and should
have a clear beginning, middle, and end.
```

**Question 1**

Does this instruction make sense and fit the task description and capability being measured?

☐ Yes

☐ No

☐ Unsure

**Question 2**

Estimate the task difficulty:

☐ Very Easy (Simple factual recall)

☐ Easy (Basic understanding, some inference)

☐ Moderate (Application of knowledge, multiple steps)

☐ Difficult (Analysis, synthesis, creative problem-solving)

☐ Very Difficult (Highly specialized knowledge, complex reasoning)

The LLM responded with:

```
Once upon a time, in a quaint little village, there lived a
young girl named Emma. Emma loved to spend her afternoons
exploring her garden, which was filled with vibrant flowers,
towering trees, and hidden nooks. One sunny day, while chasing a
 butterfly, Emma stumbled upon a curious-looking door nestled
behind a thick curtain of ivy. It was small and wooden, with
intricate carvings that seemed to shimmer in the light. With a
heart full of curiosity, Emma gently pushed the door open and
stepped through. To her amazement, she found herself in a
breathtaking world where the sky was a swirling palette of
colors, and the air was filled with the sweet scent of blooming
 flowers. Magical creatures roamed freely, from tiny fairies with
 iridescent wings to majestic unicorns that galloped across the
fields. Emma soon befriended a mischievous fairy named Lila, who
 guided her through the enchanting land. Together, they embarked
 on numerous adventures, from soaring over rainbow waterfalls to
 exploring crystal caves that sparkled with every hue. Emma felt
 a sense of wonder and joy she had never known. As the sun began
 to set, casting a golden glow over the magical world, Lila led
Emma back to the hidden door. "You must return home now," Lila
said softly, "but remember, this world will always be here for
you." With a heart full of memories, Emma stepped back through
the door and found herself in her garden once more. From that
day on, Emma's adventures in the magical world remained her most
 cherished secret. And whenever she felt the need for a bit of
magic, she knew exactly where to find it.
```

**Question 3**

Did the LLM successfully complete the task? (select an option)

☐ Yes

☐ No

☐ Unsure

---

**Question 4**
Did the LLM's success or failure on this task surprise you? (select an option)

☐ Yes

☐ No

---

## H    REPORT GENERATION

As discussed in the main paper Section 6, AUTOMATED CAPABILITY DISCOVERY can automatically produce a structured report summarizing each discovered capability and highlighting consistent successes, failures, and key insights. Below, we provide details about our workflow for generating these reports, including how clusters and tasks are automatically aggregated and summarized.

### H.1    TASK CLUSTER LABELING

After task generation, we cluster the discovered tasks (see Appendix B.3) and apply t-SNE + HDB-SCAN to group them. To label these clusters concisely, we prompt GPT-4o to summarize each cluster as follows:

---

**Task Cluster Labelling System Prompt**

You are a helpful assistant. You are given a set of tasks within a cluster.
Reply concisely and exactly in JSON format with only the following keys:
- `"thought"`: First, reason about the essence of the given tasks in the cluster.
- `"label"`: Your summary label for the cluster of tasks.
- `"capability_being_measured"`: The overall capability being measured by the tasks in this cluster.

This will be automatically parsed so ensure that the string response is precisely in the correct format.

---

**Task Cluster Labelling User Prompt**

**[DATA]**
Cluster {`cluster_id`} tasks:

Name: {`name_of_task1`}
Description: {`description_of_task1`}
Capability: {`capability_being_measured1`}

Name: {`name_of_task2`}
Description: {`description_of_task2`}
Capability: {`capability_being_measured2`}

... (any additional tasks in the cluster) ...

**[INSTRUCTION]**
Consider the above tasks in this cluster. Please provide a concise label (a natural language phrase within 10 words) for the cluster. Ensure that the label is very specific to the tasks; avoid being general. Avoid including general terms such as "problem-solving". Include more specific keywords from the tasks, such as "poem", "logic puzzles", etc.
Also, provide the overall capability being measured by the tasks in this cluster.
Return your answer as valid JSON with only the keys `"thought"`, `"label"`, and `"capability_being_measured"`.

---

These labels are then used to form summaries of the discovered tasks in our final analysis.

### H.2    REPORT GENERATION PROMPTS

Below are the prompt templates used for generating the analysis sections in the final report. This complements the discussion in the main paper Section 6.

### H.2.1 CLUSTER ANALYSIS PROMPTS

---

**Cluster Analysis System Prompt**

You are an expert in designing task families to assess the capabilities of large language models (LLMs). You will write an analytical section for a report examining the capabilities and limitations of large language models.

Your goal is to analyze and synthesize insights about LLM capabilities by examining: 1) The LLM's performance and solutions on tasks designed to test specific capabilities. 2) Any patterns, strengths, or limitations revealed through this analysis. Focus on identifying surprising successes and failures from the point of view of an expert human evaluator.

You will be given a cluster of related task families that evaluate specific LLM capabilities, along with the LLM's responses and performance on these tasks.

Your goal is to: 1) Carefully examine the example tasks and the LLM's responses 2) Analyze the LLM's proficiency level on the evaluated capabilities 3) How these examples provide meaningful insights about the model's capabilities or limitations 4) Draw meaningful conclusions about the LLM's strengths and limitations in this capability area

Respond precisely in the following format including the JSON start and end markers:

**THOUGHT**: `<THOUGHT>`
**RESPONSE JSON**: `<JSON>`

In `<THOUGHT>`, first deeply think and reason about the patterns and insights revealed by examining this cluster of related tasks.

In `<JSON>`, provide a JSON response with the following fields:
- `"overall_analysis"`: A brief conclusion based on examining the example tasks and the LLM's responses, including key capabilities demonstrated and limitations revealed
- `"surprising_example_analysis_X"`: Analysis of why this success or failure was surprising and what it reveals about the LLM's capabilities or limitations (one such field per example)
- `"insights"`: Key insights and takeaways about the LLM's capabilities based on analyzing this cluster of related tasks

For EACH provided example, include a `"surprising_example_analysis_X"` field in the JSON response, where X is replaced with the example's index number. This will be automatically parsed so ensure that the string response is precisely in the correct format.

---

**Cluster Analysis Prompt**

**Task Cluster Analysis**
Cluster Name: {cluster_name}
Capabilities Being Evaluated
{capabilities}
*Note: Please examine the examples carefully to verify which capabilities are actually being tested.*
Tasks in Cluster
{task_names}
Performance Statistics
Overall Success Rate: {overall_success_rate}
Success Rate by Task Difficulty: {difficulty_breakdown}
**Surprising Example**
Below are examples where the LLM succeeded or failed on tasks that reveal its capabilities or limitations.
{surprising_examples}
Please analyze:
1. What specific capabilities were demonstrated or lacking in the examples
2. Any patterns in the successes and failures
3. Notable or surprising results that reveal insights about the LLM's abilities
4. What this suggests about the LLM's understanding and limitations
5. How these insights connect to broader questions about LLM capabilities

### H.2.2 EXAMPLE SELECTION PROMPTS

---

**Example Selection System Prompt**

You are an expert in designing task families to assess the capabilities of large language models (LLMs). You will write an analytical section for a report examining the capabilities and limitations of large language models.
Your goal is to analyze and synthesize insights about LLM capabilities by examining:
1. The LLM's performance and solutions on tasks designed to test specific capabilities.
2. Any patterns, strengths, or limitations revealed through this analysis.
Focus on identifying surprising successes and failures from the point of view of an expert human evaluator.
You will be given a cluster of related task families that evaluate specific LLM capabilities, along with the LLM's responses and performance on these tasks. Your goal is to identify surprising successes and failures that reveal meaningful insights about LLM capabilities.
Respond precisely in the following format including the JSON start and end markers:

**THOUGHT**: `<THOUGHT>`
**RESPONSE JSON**: `<JSON>`

In `<THOUGHT>`, carefully analyze which examples demonstrate unexpected or notable behavior. Consider:
1. Surprising successes on challenging tasks that demonstrate unexpected capabilities
2. Unexpected failures on seemingly simple tasks that reveal limitations
3. Examples that challenge common assumptions about LLM capabilities
In `<JSON>`, provide a JSON response with the following fields:
- `"surprising_success_example_idx"`: List of indices for the most surprising or noteworthy successful tasks (0-3 indices)
- `"surprising_failure_example_idx"`: List of indices for the most surprising or noteworthy failed tasks (0-3 indices)
Format for index lists: Empty list `[]`, single index `[1]`, or multiple indices `[0, 1, 3]`. This will be automatically parsed so ensure that the string response is precisely in the correct format.

---

**Example Selection Prompt**

**Task Cluster Analysis**
Cluster Name: {cluster_name}
Capabilities Being Evaluated
{capabilities}
**Tasks Overview**
Total Tasks: {num_tasks}
Overall Success Rate: {overall_success_rate}
**Task Examples**
{task_examples}
Please analyze these examples carefully to identify:
1. Which examples show surprising or unexpected successes, particularly:
   - Complex tasks handled with sophisticated reasoning
   - Challenging edge cases solved successfully
   - Tasks requiring capabilities not typically associated with LLMs
2. Which examples show surprising or unexpected failures, particularly:
   - Simple tasks that unexpectedly failed
   - Inconsistent performance on similar tasks
   - Failures that reveal interesting limitations
Focus on examples that would be genuinely surprising to an LLM expert researcher and provide meaningful insights about the model's capabilities or limitations.
In your response, briefly reason about EACH provided example and explain why it is (or isn't) surprising from the perspective of an LLM expert researcher.

---

### H.2.3 OVERALL SUMMARY PROMPTS

---

**Overall Summary System Prompt**

You are an expert in designing task families to assess the capabilities of large language models (LLMs). You will write an analytical section for a report examining the capabilities and limitations of large language models.
Your goal is to analyze and synthesize insights about LLM capabilities by examining:
  1. The LLM's performance and solutions on tasks designed to test specific capabilities.
  2. Any patterns, strengths, or limitations revealed through this analysis.
Focus on identifying surprising successes and failures from the point of view of an expert human evaluator.
You are an expert researcher and engineer in Language Models. You are writing a very professional technical report to inform readers about the summary of the tested LLM's capabilities and limitations. You will now provide an overall analysis and summary of the LLM's capabilities based on all the surprising tasks identified across various clusters. Your goal is to synthesize insights about the LLM's strengths and limitations, referencing specific results from the clusters using "#Cluster_i" to refer to examples.
Respond precisely in the following format including the JSON start and end markers:

**THOUGHT**: `<THOUGHT>`
**RESPONSE JSON**: `<JSON>`

In `<THOUGHT>`, deeply analyze the patterns observed across all clusters, considering both the surprising successes and failures. Your analysis should be detailed and reference specific results, using "#Cluster_i" to refer to examples from clusters.
In `<JSON>`, provide a JSON response with the following fields:
  • `"abstract"`: An abstract to this report. The first sentence should introduce the use of the {scientist} model as a scientist to study the {subject} model's capabilities. Then summarize the main contents.
  • `"overall_summary"`: A comprehensive summary of the LLM's capabilities based on your analysis. Introduce the context for the reader, e.g. start with sentences like "In this report, we examine this LLM's . . . The LLM shows . . ."
  • `"insight"`: A very detailed and long analysis to elaborate the above summary. Be very specific. Should be a list of `str`.
  • `"surprising_capabilities"`: Key surprising capabilities demonstrated by the LLM. Should be a list of `str`, and the analysis should be detailed and long.
  • `"surprising_failures"`: Notable limitations or failures revealed. Should be a list of `str`, and the analysis should be detailed and long.
  • `"data_insights"`: Analysis and interpretation of the numerical data provided (e.g. success rates, performance statistics). Should be a list of `str`, and the analysis should be detailed and long.
This will be automatically parsed so ensure that the string response is precisely in the correct format.

---

**Overall Summary Prompt**

**Overall Summary**
You have analyzed the LLM's performance across multiple task clusters and identified surprising successes and failures.
**Scientist and Subject**
You are now using the {scientist} model as a scientist to study the {subject} model's capabilities.
**Cluster Summaries**
{cluster_summaries}
**Overall Statistics**
{overall_statistics}
Please synthesize a comprehensive analysis of the LLM's capabilities based on the information above.
In your analysis:
  1. Refer to specific results from clusters using "#Cluster_i" to refer to examples.
  2. Provide detailed observations about patterns in the LLM's performance across different clusters.
  3. Highlight surprising capabilities that challenge established understanding of LLM behavior.
  4. Discuss surprising failures that reveal significant limitations.
  5. Include analysis of numerical data, such as success rates and performance statistics.
In your response `<THOUGHT>`, provide a detailed reasoning process that leads to your conclusions.
After your analysis, provide the JSON response with the required fields.

---

## H.3 GENERATED REPORT EXAMPLE

Here we provide the first few pages of the generated report by AUTOMATED CAPABILITY DISCOVERY on GPT-4o (serving as both scientist and subject), as described in Section 6. Please find full reports for all evaluation settings in Section 5 at `https://github.com/conglu1997/ACD/tree/main/reports`.

# ACD Capability Report on GPT-4o Subject

**GPT-4o**

## ABSTRACT

Using the GPT-4 model as both scientist and subject, this report examines the capabilities and limitations of the GPT-4 model across various task clusters. By analyzing its performance, we identify both surprising successes and notable failures, offering insights into its proficiency in procedural tasks, scientific reasoning, legal analysis, and more. The report synthesizes these findings to highlight the model's strengths and areas for improvement, providing a comprehensive overview of its potential applications and limitations.

## Contents

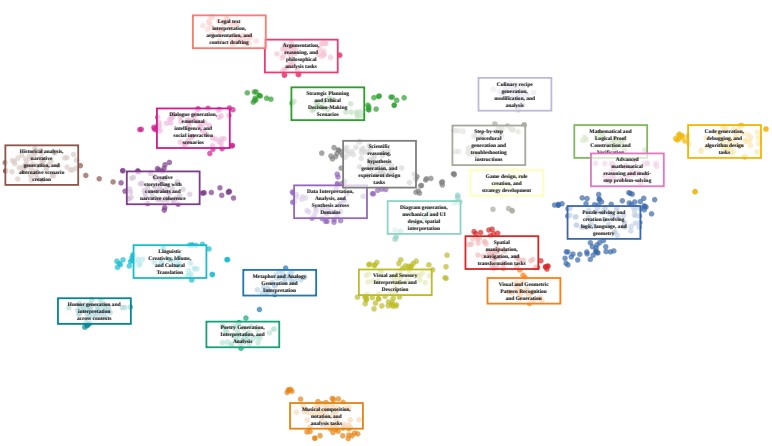

**Figure 1.** Visualization of task families discovered by ACD on GPT-4o (serving as both *scientist* and *subject*) over 5000 generations.

## 1 Overview

In this report, we are going to examine this LLM's capabilities and limitations across various task clusters. The LLM shows strong performance in structured tasks requiring procedural understanding, legal reasoning, and scientific communication. However, it faces challenges in dynamic and abstract problem-solving scenarios, such as advanced mathematical reasoning and strategic planning. These findings highlight the model's strengths in specific domains while pointing to areas needing further enhancement.

### 1.1 Insights

- The LLM excels in tasks requiring procedural understanding and technical communication, particularly in Step-by-step procedural generation and troubleshooting instructions, where it achieves a high success rate in tasks like origami instructions, demonstrating strong spatial reasoning and instructional clarity.
- In Scientific reasoning, hypothesis generation, and experiment design tasks, the model shows proficiency in scientific reasoning and simplifying complex concepts, although it struggles with experimental design for abstract phenomena, indicating a need for improved operationalization of scientific ideas.
- The model's legal reasoning and document generation capabilities are highlighted in Legal text interpretation, argumentation, and contract drafting, where it effectively interprets legal texts and constructs arguments, suggesting its utility in legal research and document preparation.
- Despite strengths in structured reasoning, the LLM struggles with dynamic and strategic tasks, as seen in Game design, rule creation, and strategy development, where it fails in complex pathfinding and chess strategy tasks, pointing to limitations in spatial reasoning and domain-specific adaptations.
- The analysis of numerical data reveals high success rates in clusters involving scientific reasoning and historical analysis, suggesting strong interdisciplinary synthesis capabilities, but highlights weaknesses in advanced mathematical reasoning, indicating areas for improvement.

### 1.2 Surprising Capabilities

- The LLM's ability to generate coherent step-by-step instructions in Step-by-step procedural generation and troubleshooting instructions, particularly for tasks like origami, showcases a surprising proficiency in spatial reasoning and procedural

**2**

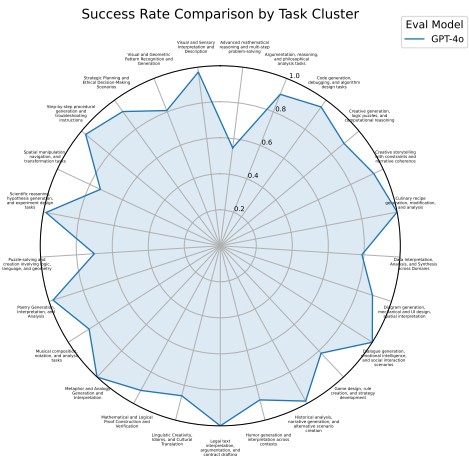

**Figure 2.** Success rates on each cluster of tasks.

communication, suggesting potential applications in education and technical writing.
- In Scientific reasoning, hypothesis generation, and experiment design tasks, the model's capability to simplify complex scientific concepts into accessible explanations demonstrates a notable strength in scientific communication, although with limitations in experimental design.
- The high success rate in legal reasoning tasks in Legal text interpretation, argumentation, and contract drafting reveals a surprising depth of understanding in legal principles and the ability to generate coherent legal documents, highlighting its utility in legal domains.

### 1.3 Surprising Failures
- The LLM's inability to effectively handle dynamic and strategic reasoning tasks, as evidenced in Game design, rule creation, and strategy development, where it struggles with pathfinding and chess strategy, indicates a significant limitation in adapting to dynamic environments and integrating spatial considerations.
- In Advanced mathematical reasoning and multi-step problem-solving, the model's lower success rate in advanced mathematical reasoning tasks, including complex mathematical modeling and symbolic manipulation, reveals a critical shortcoming in its mathematical understanding and problem-solving capabilities.
- Despite strengths in abstract reasoning, the model's performance in Mathematical and Logical Proof Construction and Verification, where it shows weaknesses in generating basic mathematical proofs, suggests an inconsistency in logical reasoning across different complexity levels.

### 1.4 Data Insights
- The overall success rate of 87.57% indicates strong performance across many clusters, yet significant variability suggests certain domains where the model excels versus those it struggles with.
- Clusters with the highest success rates, such as Scientific reasoning, hypothesis generation, and experiment design tasks (97.75%) and Historical analysis, narrative generation, and alternative scenario creation (97.50%), highlight the model's proficiency in interdisciplinary reasoning and historical analysis, suggesting effective synthesis and creative capabilities.
- The notably lower success rate in Advanced mathematical reasoning and multi-step problem-solving (56.13%) underscores the LLM's limitations in handling complex mathematical tasks, pointing to an area that requires further enhancement and training.
- The success rates across clusters reveal a pattern where the model performs well in structured and rule-based tasks but faces challenges in dynamic, strategic, and abstract problem-solving scenarios.

**3**

## 2 Detailed Task Analysis

### 2.1 Step-by-step procedural generation and troubleshooting instructions

#### 2.1.1 Overview

**Capabilities**: Procedural understanding, technical communication, and instructional clarity

**Number of Tasks**: 34

**Success Rate**: 92.94%

**Difficulty Success Rates**: - moderate: 95.00% - hard: 91.50%

**Difficulty Percentages**: - moderate: 41.2%

- hard: 58.8%

#### 2.1.2 Analysis

The LLM demonstrates strong procedural understanding and technical communication skills, particularly in tasks requiring detailed step-by-step instructions and spatial reasoning, as reflected by the high success rates in origami-related tasks.

**Insights**:

The model excels in tasks demanding clear procedural instructions and spatial understanding, such as origami, revealing strengths in tasks that require detailed sequencing and visualization. This capability suggests potential applications in technical writing and documentation fields, although it may face challenges in more abstract or less structured procedural tasks.

#### 2.1.3 Task Examples

**Example 1**

**Task**: origami_instructions **Task Description**: Interpret given origami instructions and generate new origami instructions based on specified shapes. **Difficulty Level**: 4 (hard) **Instructions**: Your task is to interpret the following origami instructions and describe the resulting shape. Provide your description in plain text format.

Instructions: Fold a square paper in half diagonally to form a triangle. Unfold, then fold the paper in half diagonally in the opposite direction to form a triangle. Unfold, then fold the paper in half horizontally to form a rectangle. Unfold, then fold the paper in half vertically to form a rectangle. Unfold.

Ensure your description is clear and accurately represents the final shape formed by following the instructions. **Model Response Example**: The resulting shape is a square piece of paper with creases forming an asterisk (*) pattern. The creases intersect at the center, dividing the square into eight triangular sections. **Judge for Above Example**: Successful **Overall Success Rate**: 100.0%

The LLM's ability to accurately interpret and describe the final shape from origami instructions was surprising due to the spatial reasoning required. This success indicates a robust capacity for understanding and visualizing procedural instructions, suggesting potential strength in similar complex procedural tasks.

**Example 2**

**Task**: origami_instructions_generation **Task Description**: Generate and interpret step-by-step origami instructions to create specific origami models. **Difficulty Level**: 4 (hard) **Instructions**: Your task is to generate step-by-step origami instructions to create the specified origami model. Begin with the given initial step and ensure that each step is clear, detailed, and necessary. The final step should complete the origami model. Here is the information about the task:

Goal Model: crane Initial Step: Start with a square piece of paper with the colored side up.

Submit your instructions in plain text format. Use the following format for each step: Step 1: [Description of step] Step 2: [Description of step] … Final Step: [Final description completing the model]

**Model Response Example**: Step 1: Start with a square piece of paper with the colored side up. Step 2: Fold the paper in half diagonally to form a triangle, then unfold it. Step 3: Fold the paper in half diagonally in the other direction to form a triangle, then unfold it again. You should now have two creases forming an "X." Step 4: Turn the paper over so the white side (or the side without color) is facing up. Step 5: Fold the paper in half horizontally to form a rectangle, then unfold it. Step 6: Fold the paper in half vertically to form a rectangle, then unfold it. You should now have a grid of creases. Step 7: Collapse the paper into a square base by bringing the horizontal and vertical creases together, forming a smaller square with the open ends at the bottom. Step 8: Rotate the square base so that it looks like a diamond with the open ends pointing down. Step 9: Fold the right corner of the top layer to the center crease, then fold the left corner of the top layer to the center crease. The top layer should now have a

**4**

