# OpenReview forum: "Automated Capability Discovery via Model Self-Exploration"
_ICLR.cc/2025/Workshop/BuildingTrust — BuildingTrust_

### Official Review · Reviewer_r6aF · 2025-02-24

**Rating:** 6
**Confidence:** 3

**Review:**

Summary:  This paper proposes Automated Capability Discovery (ACD), a framework that systematically proposes open-ended evaluation tasks to discover new capabilities and failures of models.

Strengths:
- The problem of model evaluation has become increasingly more important and difficult, particularly as people begin to interact with models in unexpected and potentially dangerous ways. Current evaluations are often quite ad-hoc and limited, and this paper proposes an interesting approach to induce more structure in coming up with new tasks for evaluations and discovering failures and capabilities of new models.
- The proposed framework is very flexible and provides an interesting approach to scalable oversight, where human oversight is aided by models that are good at /proposing/ tasks, even if they are incapable of performing them themselves.

Weaknesses:
- The evaluation of the proposed automated discovery approach is a bit lacking at the moment. In particular, it isn't clear whether the authors were able to derive new conclusions about models that have not or could not be found with existing evaluation or human-based interaction approaches. Essentially, how does the method compare to the existing baseline of human evaluations? More discussion on this point would be very beneficial.
- There was no verification of the validity of the "interesting and new" metric the authors used to evaluate the diversity of the discovered tasks. It would have been interesting to include this in the human survey conducted to verify that the tasks were actually novel with respect to each other.
- It is unclear whether the discovered tasks and capabilities are core, intuitive tasks, or if the method was truly able to uncover fringe, unexpected behavior of the models. Were the authors able to find any safety-critical failures of models or any completely unexpected abilities that weren't previously known?

---

### Official Review · Reviewer_9FBa · 2025-02-28

**Rating:** 7
**Confidence:** 3

**Review:**

Summary

The paper introduces Automated Capability Discovery (ACD), a novel framework that leverages foundation models such as such as GPT, Claude, and Llama, to evaluate themselves or other models by autonomously generating and testing new tasks. ACD choses one model as a scientist to create open-ended challenges for another subject model, mimicing on manually designed benchmarks with human-trial-and-error. ACD iteratively probes the capabilities and weaknesses of the subject model, enabling scalable and systematic evaluation across reasoning, math, code generation, and creative problem-solving.
Beyond detecting failure modes, ACD also facilitates cross-model comparisons by curating a repository of tasks that can be used to benchmark different AI systems under consistent evaluation conditions. The results show that different scientist models generate distinct probing challenges, suggesting that an ensemble approach could enhance capabilities coverage. The authors argue that ACD is a step toward scalable and continuous AI evaluation, helping improve model reliability and safety by surfacing potential risks before deployment.

Review

The paper addresses a crucial challenge in AI evaluation by proposing an automated framework that scales beyond human-designed benchmarks. ACD effectively leverages the generative capabilities of foundation models to explore their own strengths and weaknesses, reducing human effort while increasing coverage of model capabilities and failures. The results are compelling, demonstrating ACD’s ability to generate meaningful tasks and validate its findings with human evaluations. The ability to use different scientist models for task generation is also a good contribution, as it shows that diverse models can uncover different aspects of AI performance. Overall, I think this is a well-executed study that offers both efficiency and depth in AI model capability discovery. While there are areas for future improvement, e.g., extend to multi-modal models, the ACD framework represents a meaningful advancement in AI model assessment.

---

### Official Review · Reviewer_WK1A · 2025-03-01
**ACD contributes needed innovation to the field of evals; room for more clarity**

**Rating:** 8
**Confidence:** 4

**Review:**

Quality: 8
- Pros: authors make a meaningful contribution to evaluations, fostering innovation and acceleration amidst rapidly evolving models.
- Cons/areas for improvement: a) Authors might consider clearly stating what evaluation objectives ACD can accomplish and which it cannot (i.e. can it be used for benchmarking capabilities between models? can it be used to probe specific dangerous capabilities? are there any tradeoffs of its exploratory nature?) b) Figure 1 does’t accurately represent the status quo evaluation alternative to ACD — the human counterfactual is quite reductive and simplistic and there are in fact options between this and ACD. c) Other minor issues: Gpt-4o is repeated twice in line 1128, should line 939 read "you should *not* use this function"?

Clarity: 7
- Pros: graphics, description, and supplementary materials portray process clearly.
- Cons/areas for improvement: a) More information could also be provided on the humans in the large-scale human surveys (i.e. demographics, country, education, age, etc.). b) Clarify where human decision-making comes into play in ACD's process. How does the scientist model know if it requires multiple refinements (line 984)? Can users request that ACD investigate particular task families? c) Clarify whether the judge always scores binarily and, if so, how this decision was made (i.e. relative to a more granular, continuous score). Does the judge respond differently if the system prompt is slightly tweaked? How is the criteria created for the LLM judge and what limitations might exist? d) Clarify whether this is for foundation models more generally (e.g. different modalities) or just LLMs. Both terms are used.

Originality: 9
- Pros: dynamic/automated evaluations are underexplored and this is a useful contribution.
- Cons: Other work has been done on autograders; the authors could acknowledge the range of variation in LLM-as-a-judge, explain their design decisions for this particular judge, limitations to the particular LLM-as-a-judge approach, and variations to explore in future research.

Significance: 7
- Pros: see "quality"
- Cons: to give context to the significance of this piece, the authors could provide more information on comparisons between the performance/abilities of ACD relative to other similar tools in the field. In what ways does it outperform other evaluation tools and it what ways does it fall short? b) D1 shows that new tasks emerge but do the new tasks give us sufficiently new information about the models and real-world capabilities? i.e. how does variance of tasks change over generation?

---

### Decision · Program_Chairs · 2025-03-04

**Decision:**

Accept

**Comment:**

The reviewers agree that ACD is a well-executed and timely approach to tackling the increasingly difficult problem of evaluating foundation models. The paper is a good fit for the workshop, as model evaluation is crucial for building trust in AI systems. The flexibility of ACD in leveraging different "scientist" models and its ability to uncover previously unknown behaviors further enhance its relevance. It would have been nice to see discussion on whether ACD uncovers safety-critical failures or merely expected model behavior.